# Contrasting roles for parvalbumin-expressing inhibitory neurons in two forms of adult visual cortical plasticity

Eitan S Kaplan[1†], Sam F Cooke[1†], Robert W Komorowski[1], Alexander A Chubykin[2], Aurore Thomazeau[1], Lena A Khibnik[3], Jeffrey P Gavornik[4], Mark F Bear[1*]

[1]Picower Institute for Learning and Memory, Massachusetts Institute of Technology, Cambridge, United States; [2]Department of Biological Sciences, Purdue University, West Lafayette, United States; [3]Department of Neurology, Sanford Health, Fargo, United States; [4]Department of Biology, Boston University, Boston, United States

**Abstract** The roles played by cortical inhibitory neurons in experience-dependent plasticity are not well understood. Here we evaluate the participation of parvalbumin-expressing (PV+) GABAergic neurons in two forms of experience-dependent modification of primary visual cortex (V1) in adult mice: ocular dominance (OD) plasticity resulting from monocular deprivation and stimulus-selective response potentiation (SRP) resulting from enriched visual experience. These two forms of plasticity are triggered by different events but lead to a similar increase in visual cortical response. Both also require the NMDA class of glutamate receptor (NMDAR). However, we find that PV+ inhibitory neurons in V1 play a critical role in the expression of SRP and its behavioral correlate of familiarity recognition, but not in the expression of OD plasticity. Furthermore, NMDARs expressed within PV+ cells, reversibly inhibited by the psychotomimetic drug ketamine, play a critical role in SRP, but not in the induction or expression of adult OD plasticity.

*For correspondence: mbear@mit.edu

†These authors contributed equally to this work

Competing interests: The authors declare that no competing interests exist.

## Introduction

Understanding how brain synapses, cells, and circuits are persistently modified by experience to store information represents one of the great challenges in neuroscience. Mouse visual cortex has proven to be an excellent model system in which to examine experience-dependent neural response modification. One robust type of visual response plasticity is reliably elicited in adult (> P60) mice by simply closing one eyelid. Over the course of 5–7 days of monocular deprivation (MD), the responses in visual cortex evoked by stimulation of the non-deprived eye progressively increase (*Sawtell et al., 2003*; *Sato and Stryker, 2008*). This deprivation-enabled response potentiation is driven by visual experience through the non-deprived eye, as only the responses through one eye are potentiated (*i. e.*, it is input specific) and it fails to occur if both eyelids are closed or if the animals are kept in a dark room (*Blais et al., 2008*). There is evidence that the response potentiation is mediated in part by "Hebbian" strengthening of excitatory synaptic transmission in visual cortex, as induction requires cortical NMDA receptor (NMDAR) activation (*Sawtell et al., 2003*) (*Sato and Stryker, 2008*) and α-calcium/calmodulin-dependent protein kinase II (αCAMKII) expression in principal cells (*Ranson et al., 2012*). This form of ocular dominance (OD) plasticity is likely responsible for the increase in visual acuity that occurs through the non-deprived eye following adult monocular deprivation (*Iny et al., 2006*), and is of particular interest in the context of recovery of brain function after deprivation, disease, or damage (*Cho and Bear, 2010*).

**eLife digest** What we see or fail to see through our eyes leaves a lasting impression by changing the strength of connections between neurons in a part of the brain called the visual cortex. These changes are referred to as synaptic plasticity.

One example of synaptic plasticity results in the visual cortex becoming more responsive to the stimulation of one eye when the other eye is patched for about a week. This phenomenon is known as "ocular dominance plasticity". Another example is the increase in responsiveness that occurs in the visual cortex when animals repeatedly view stripes of a single orientation. This phenomenon is known as "stimulus-selective response potentiation". Some previous studies had suggested that both kinds of plasticity might be induced in the same way. However, both forms of plasticity can happen at the same time, suggesting that distinct mechanisms may be involved.

To tease out how these two kinds of plasticity work, Kaplan, Cooke et al. inactivated one particular type of neuron that is thought to be involved in triggering ocular dominance plasticity and is found in the visual cortex. These inhibitory neurons produce a molecular marker called parvalbumin and are therefore referred to as "parvalbumin-expressing neurons".

The experiments showed that ocular dominance plasticity could still be seen in adult mice when these parvalbumin-expressing neurons were inactivated. However, when these same neurons were inactivated, the visual cortex no longer responded differently to lines with familiar or new orientations. This was the case even when the mice had seen lines in a given orientation for long periods of time. Similarly, observations of the behavior of the mice also showed that their ability to distinguish new from familiar stimuli was lost when the parvalbumin-expressing neurons were inactivated locally within part of the visual cortex.

The connections between neurons that bring information from the eyes to the visual cortex release a chemical neurotransmitter called glutamate. One important protein that detects glutamate, called an NMDA receptor, is required for ocular dominance plasticity. Further experiments showed that NMDA receptors within the parvalbumin-expressing neurons are needed for stimulus-selective response potentiation, but not for ocular dominance plasticity. This indicates that although the two forms of plasticity may share molecular requirements, different connections and cells types are likely involved.

Changes in parvalbumin-expressing neurons and NMDA receptors have been implicated in disorders such as autism and schizophrenia. To improve treatments for these disorders, it is crucially important to better understand the links between these neurons and the retrieval of memories.

Another robust form of visual response plasticity is induced by exposure of awake mice to oriented visual grating stimuli. Brief daily presentation of a phase-reversing grating of a single orientation causes a large and persistent increase in the peak cortical response to this orientation, as measured by visual evoked potentials (VEPs) or unit recordings (*Frenkel et al., 2006*; *Cooke et al., 2015*). This phenomenon is termed stimulus-selective response potentiation (SRP) because only responses to the experienced orientation are increased. Abundant evidence suggests that SRP is also mediated by "Hebbian" mechanisms, particularly those revealed by the study of long-term synaptic potentiation (*Cooke and Bear, 2010*; *2014*). The mechanisms utilized for SRP within V1 have also been shown to mediate a fundamental form of long-term visual recognition memory, manifested behaviorally as orientation-selective habituation (OSH) (*Cooke and Bear, 2015*; *Cooke et al., 2015*).

Both monocular deprivation and selective visual experience trigger input-specific increases in the short latency VEP measured in layer 4 of mouse visual cortex and, as reviewed above, both OD plasticity and SRP share some molecular requirements (*e.g.,* NMDAR activation). Thus, it came as a surprise that response potentiation after monocular deprivation and SRP do not occlude one another (*Frenkel and Bear, 2004*), suggesting that they employ different mechanisms or are expressed by different synapses. We became interested in the possibility of differential involvement of cortical inhibition mediated by the parvalbumin-expressing (PV+) fast spiking neurons. PV+ fast-spiking inhibitory neurons comprise the most numerous sub-class of GABAergic cortical neurons (*Xu et al., 2010*) and receive substantial feed-forward glutamatergic input from the thalamus

(*Cruikshank et al., 2007*). These interneurons synapse preferentially on the somata and initial axon segments of principal cells, and therefore are in a position to strongly and precisely modulate even short latency visually evoked responses (*Cristo et al., 2004*; *Markram et al., 2004*; *Kepecs and Fishell, 2014*). Moreover, previous studies have suggested that fast-spiking interneurons participate in the expression of cortical eye dominance and OD plasticity in juvenile mice (*Yazaki-Sugiyama et al., 2009*; *Smith and Bear, 2010*). The contribution of this class of neuron to cortical plasticity is also of particular interest given the evidence for cortical PV+ neuron dysfunction in various psychiatric disorders characterized by impaired cognitive function (*Gogolla et al., 2009*).

In the current study we used a variety of approaches to understand the contribution of PV+ neurons to adult OD plasticity and SRP. We find that neither the pharmacogenetic silencing of PV+ cells nor the specific deletion of NMDARs within them affect the relative eye dominance or the expression of deprivation-enabled potentiation of VEPs in adult mice. In contrast, and quite unexpectedly, we discovered that the expression of SRP is dependent on PV+ cell activity, and that perturbations of PV+ neuron function, most notably cell-type specific ablation of NMDARs, disrupt the expression of both SRP and its behavioral correlate, familiarity recognition.

## Results

### Ocular dominance is maintained in the absence of PV+ neuron activity

In order to understand the role of PV+ neurons in the expression of experience-dependent visual cortical plasticity, we selectively inactivated these cells using a Dreadd (Designer Receptors Exclusively Activated by Designer Drugs) pharmacogenetic system: Specifically, we expressed a re-engineered G-protein coupled receptor, hM4D(Gi), which is activated exclusively by an otherwise inert small molecule, clozapine-N-oxide (CNO) (*Nichols and Roth, 2009*). The binding of CNO to the hM4D(Gi) receptor activates intracellular $G_i$-mediated signaling and subsequent hyperpolarization of the cell in which the receptors are expressed. We locally expressed hM4D(Gi) receptors in PV+ cells of binocular V1 using an adeno-associated viral vector containing a construct for Cre-selective expression (AAV9-hSyn-DIO-HA-hM4D(Gi)-IRES-mCitrine) in mice that express Cre recombinase only in PV+ cells (B6;129P2-$Pvalb^{tm1(cre)Arbr}$/J, PV-Cre). We confirmed the selective expression of hM4D(Gi) receptors in PV+ neurons in binocular V1 using immunohistochemistry (*Figure 1A–E*). In order to ensure that PV+ neurons could be inactivated by this method, we took slices of V1 for ex vivo intracellular recordings. Bath application of CNO drastically inhibited current evoked action potential firing of PV+ neurons in Layer 4 (2-way repeated measures ANOVA, interaction of CNO x current injection, $F_{(8,72)} = 6.227$, $P < 0.001$, n = 10 cells, significant at data points above 100pA: $q(8) = 3.716$, $P = 0.014$) but had no effect on neighboring non-expressing cells (*Figure 1F–G*). Layer 4 recordings in vivo demonstrated that systemic administration of CNO resulted in the elevation of visually-evoked potential (VEP) magnitude, and an increase in the firing rate of excitatory single units (*Figure 1H–I*), which are expected consequences of inactivating PV+ inhibitory neurons in binocular V1.

Mouse binocular V1 is ~2–3 times more strongly activated by the contralateral (contra) eye than the ipsilateral (ipsi) eye, an inherent bias known as ocular dominance (OD). We probed the involvement of PV+ neuron activity in maintaining baseline OD. PV-Cre mice were infected with AAV virus to deliver hM4D(Gi) Dreadd receptors to PV+ neurons in binocular V1 (*Figure 2A*). Electrodes were simultaneously implanted into layer 4 of V1 for VEP recordings. VEPs elicited through just contralateral or ipsilateral eyes were acquired consecutively before and after CNO administration (*Figure 2A–B*). After CNO injection, VEPs driven through each eye increased significantly in magnitude (2-way repeated measures ANOVA, effect of CNO, $F_{(7)} = 75.986$, $P < 0.001$; contralateral eye: 222.18 ± 14.79 μV baseline vs. 679.81 ± 64.12 μV CNO, Student-Newman-Keuls (SNK) post hoc test, $q_{(7)} = 13.925$, n = 8 mice, $P < 0.001$; ipsilateral eye: 105.56 ± 6.78 μV baseline vs. 358.63 ± 35.79 μV CNO, SNK post hoc test, $q_{(7)} = 7.711$, n = 8 mice, $P < 0.001$, (*Figure 2C*), consistent with the occurrence of cortical disinhibition. However, the OD of visual responses in V1 (contra/ipsi ratio) was not significantly altered by CNO injection (2.13 ± 0.12 baseline vs. 1.96 ± 0.18 CNO, student's paired two-tailed t-test, $t_{(7)} = 0.859$, n = 8 mice, $P = 0.42$, *Figure 2D*). This result suggests that the inherent OD of binocular visual cortex is not dependent on PV+ neuron activity.

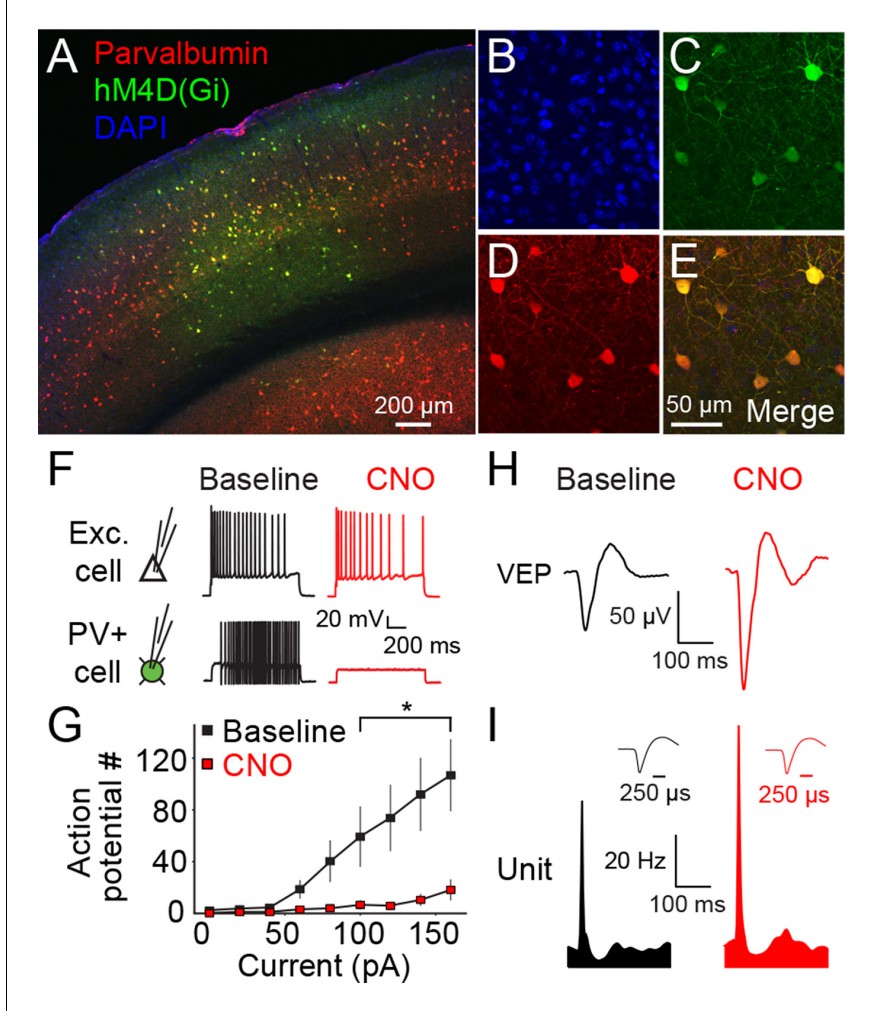

**Figure 1.** The hM4D(Gi) DREADD system locally inactivates parvalbumin+ neurons in binocular primary visual cortex (V1). (**A**) An example of V1 expression of hM4D(Gi) in a parvalbumin (PV)-Cre recombinase (Cre) mouse infected locally in binocular V1 with AAV9-hSyn-DIO-hM4D(Gi)-mCitrine. (**B**) A DAPI stain for cell nuclei is shown in blue. (**C**) Infected cells expressing hM4D(Gi) are labeled in green. (**D**) Immuno-labeled PV+ cells are shown in red. (**E**) The merged image reveals that hM4D(Gi)-expressing cells are also PV+. (**F**) Intracellular current clamp recordings of hM4D(Gi)-infected PV+ layer 4 neurons in *ex vivo* slices of V1 reveal that green-labeled infected cells exhibit a non-adapting fast-spiking phenotype typical of fast-spiking inhibitory neurons (black). These cells do not fire action potentials in the presence of CNO (red), the exogenous ligand for hM4D(Gi) receptors, despite depolarizing current injection. In contrast, neighboring cells that are not mCitrine+ show no impact of CNO application. (**G**) HM4D(Gi)-mediated inactivation of putative fast-spiking PV+ inhibitory neurons is here summarized as the number of action potentials resulting from a given current injection before (black) and after CNO application (red). (**H**) The effects of hM4D(Gi)-mediated inactivation of putative PV+ fast-spiking inhibitory neurons *in vivo* are apparent from electrophysiological recordings from V1 of awake, head-fixed mice viewing phase-reversing sinusoidal grating stimuli. Averaged Visually Evoked Potential (VEP) recordings recorded in layer 4 reveal increased VEP magnitude in the presence of CNO (red), relative to pre-CNO (baseline) recordings (black), indicative of reduced inhibition. (**I**) Phase reversal-evoked action potentials recorded from neurons in layer 4 also exhibit a similar effect with elevated firing rates in the presence of CNO (red) relative to the baseline recording (black). Labeled scale bars are presented throughout. Error bars are standard error of the mean (S.E.M.).

The following source data is available for figure 1:

**Source data 1.** Action potential number for current injection.

## Expression of adult ocular dominance (OD) plasticity does not require the activity of PV+ neurons

The normal OD ratio is altered in the adult mouse by MD of the contralateral eye over 7 days, resulting in an ocular dominance shift that features potentiation of response through the open ipsilateral eye (*Sawtell et al., 2003*; *Sato and Stryker, 2008*). In a new group of mice we assayed the effect of PV+ neuron inactivation on the expression of the shift in OD caused by MD. We recorded VEPs elicited through each eye from PV-Cre mice expressing hM4D(Gi) receptors in binocular V1 (*Figure 2E*). After baseline recordings the mice underwent contralateral eyelid suture and 7 days of monocular deprivation. Subsequently, the contralateral (deprived) eye was opened and VEPs were re-recorded in order to reveal the expression of an OD shift. There was a significant potentiation of VEP magnitude driven through the ipsilateral eye following 7 days of MD (69.5 ± 5.81 μV pre MD vs. 139.5 ± 9.33 μV post MD, n = 10 mice, student's paired one-tailed t-test, $t_{(9)}$ = -10.184, P < 0.001, *Figure 2F*). This potentiation of response through the ipsilateral (non-deprived) eye resulted in a significant shift in the OD ratio (contra/ipsi ratio, 3.08 ± 0.26 pre MD vs. 1.33 ± 0.10 post MD, 1-way repeated measures ANOVA, SNK post hoc test, $q_{(2)}$ = 11.77, P < 0.001, *Figure 2G*). Mice were then injected with CNO and re-recorded to assess whether the OD shift would persist in the absence of PV+ neuron activity. CNO injection resulted in an increase in VEP magnitudes (*Figure 2F*), but importantly the shift in the contra/ipsi ratio was maintained (1.33 ± 0.10 post MD vs. 1.35 ± 0.18 post MD CNO, SNK post hoc test, $q_{(2)}$ = 0.127, P = 0.93, *Figure 2G*). Therefore, the expression of the adult OD shift induced by 7 days of MD persists after a strong reduction in PV+ neuron inhibition.

## Inactivation of PV+ neurons disrupts expression of stimulus-selective response potentiation (SRP)

We next assayed the role of PV+ neurons in the expression mechanism underlying stimulus-selective response potentiation (SRP), a second form of experience-dependent visual cortical plasticity that also manifests as an increase in V1 responses (*Frenkel et al., 2006*; *Cooke and Bear, 2010*). Binocular VEPs were recorded from awake, head-fixed adult mice viewing phase-reversing gratings of a particular orientation (X° stimulus) on each of 6 consecutive days (*Figure 3A and B*). On the 7th day mice viewed blocks of the now familiar visual stimulus interleaved with blocks of a novel oriented stimulus (X +/- 60°). To address whether PV+ neuron activity is required for the expression of SRP, familiar and novel oriented grating stimuli were presented before and after mice received CNO (*Figure 3A*).

As is characteristic of SRP, there was a significant increase in the magnitude of the average VEP evoked by the familiar oriented grating stimulus over days (Friedman 1-way repeated measures ANOVA on ranks, n = 10 mice, $X^2_{(5)}$ = 40.40, P < 0.001, *Figure 3C*). This potentiation was evident by day 2 (263.3 ± 19.96 μV) in comparison with day 1 (184 ± 17.13 μV, SNK post hoc test, $q_{(9)}$ = 4.472, P < 0.05). The stimulus selectivity of VEP magnitude potentiation was apparent on day 7 and significantly affected by inactivating PV+ inhibitory neurons in V1 (2-way repeated measures ANOVA, interaction of treatment x stimulus, F = 78.927$_{(1,9)}$, P < 0.001, *Figure 3D*): Prior to delivery of CNO the average VEP magnitude driven by the familiar stimulus (324.7 ± 22.18 μV) was significantly greater than that driven by a novel oriented stimulus (162.9 ± 16.31 μV, SNK post hoc test, $q_{(9)}$ = 10.709, P < 0.001). Following injection of CNO to inactivate hM4D(Gi)-infected neurons in V1, there was no longer a significant difference in the magnitude of VEPs driven by the familiar stimulus (537.2 ± 66.69 μV) compared with a novel stimulus (525.1 ± 68.30 μV, SNK post hoc test, $q_{(9)}$ = 0.804, P = 0.58). The selectivity of the potentiation to a familiar stimulus can be summarized by plotting the ratio of VEP magnitudes driven by familiar and novel stimuli (*Figure 3E*). Mice expressed a significantly larger familiar/novel ratio before CNO injection (2.11 ± 0.17), corresponding to a greater response to the familiar stimulus, compared to after CNO injection (1.03 ± 0.04, Mann Whitney rank sum test, U = 0.00, P < 0.001). CNO injection did not affect the stimulus selectivity of SRP in wild type animals infected with virus (*Figure 3—figure supplement 1A–B*). These animals underwent SRP (*Figure 3—figure supplement 1A*) and on test day showed significantly larger VEPs to the familiar stimulus, both prior to CNO injection (familiar VEP: 361.06 ± 27.97 μV, novel VEP: 219.25 ± 17.9 μV, 2-way repeated measures ANOVA, n = 8 mice, SNK post hoc test, $q_{(7)}$ = 13.618, P < 0.001) and following CNO injection (familiar VEP: 369.00 ± 31.16 μV, novel VEP: 256.75 ± 19.59 μV, SNK

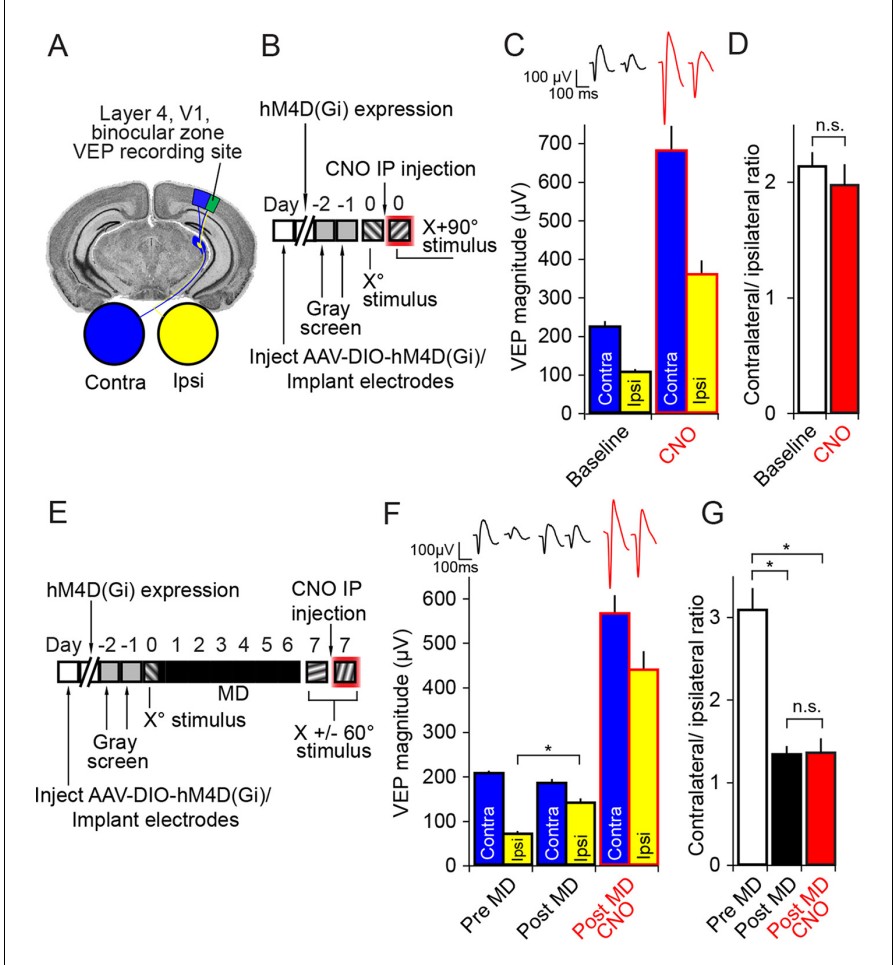

**Figure 2.** Inactivation of parvalbumin+ neurons has no impact on expression of ocular dominance (OD) or the ocular dominance shift as a result of monocular deprivation (MD) in the adult mouse. (**A**) For experiments described in this figure mice were infected across all cortical depths bilaterally in binocular V1 (green). During the same surgical implantation procedure VEP recording electrodes were also positioned in layer 4 and chronically fixed. (**B**) Mice (P45-60) were infected and implanted with electrodes and then left for 3 weeks for the AAV9 viral vector to reach maximal expression, after which they underwent habituation to head-fixation and a gray screen for two consecutive days. Following this, on experimental day 0, mice were presented with an X° phase-reversing sinusoidal grating stimulus separately to the left and right eye. VEPs were recorded from each hemisphere in order to determine ocular dominance in binocular V1. Mice were then removed from the recording apparatus and, CNO was delivered systemically half an hour prior to undertaking the same recording procedure, this time using an orthogonal X + 90° visual stimulus. (**C**) As is well documented, responses in V1 to stimuli viewed through the contralateral eye (blue) were greater in magnitude than those elicited through the ipsilateral eye (yellow), as measured here by VEP magnitude. After application of CNO (red outlines), VEP magnitude dramatically increased. (**D**) This increase was scaled such that the ratio of Contralateral:Ipsilateral VEP magnitude was maintained before (white) and after CNO (red). (**E**) In a second group of mice, a similar experimental protocol was observed prior to measuring ocular dominance by recording VEPs in each hemisphere elicited by an X° stimulus through each eye. One hemisphere was then selected and the contralateral eye was sutured closed. After 7 days of monocular deprivation, the eye was opened and VEPs driven through either eye were again recorded, this time elicited by an X + 60° stimulus. Mice were then systemically injected with CNO and, half an hour later, VEPs driven through either eye were recorded, this time elicited by an X - 60° stimulus. (**F**) After the adult mice underwent 7 days of MD, there was a significant potentiation of the V1 response to visual input through the ipsilateral eye (yellow). After application of CNO (red outlines), VEPs driven through each eye were elevated in magnitude, but again the increase in VEP magnitude was scaled. (**G**) As a result of open eye potentiation after MD, the OD ratio shifted dramatically from the contralateral bias of pre MD (white) to an almost equal cortical response through contralateral and ipsilateral eyes (black). This shifted ratio was unaffected by hM4Di-mediated inactivation of putative fast-spiking inhibitory neurons during CNO application (red), indicating that the expression of OD and its shift as a result of MD in adult mice do not require fast-spiking inhibition. Significant comparisons are labeled with an asterisk and non-significant comparisons with n.s. throughout. Error bars are standard error of the mean (S. E.M.).

The following source data is available for figure 2:

**Source data 1.** Ocular dominance and deprivation effects are maintained after PV neuron inactivation.

post hoc test, $q_{(7)}$ = 10.779, P < 0.001, *Figure 3—figure supplement 1B*). These results show that the inactivation of PV+ neurons in binocular V1 disrupts the expression of SRP.

## Disruption of SRP by PV+ neuronal inactivation is not due to saturation of responses

Cortical neurons respond within a dynamic range, and it is possible that discrimination of familiar and novel stimuli would be lost as responses approach saturation. Our observation that OD is maintained after PV+ neuron inactivation (*Figure 2*) indicates that V1 can still respond selectively to a strong (contralateral eye) and weak (ipsilateral eye) input. However, to address the possibility that a "ceiling effect" contributes to the disruption of SRP expression during PV+ cell inactivation, we conducted an additional experiment in which mice viewed sinusoidal grating stimuli across a range of contrast values (5, 10, 25, 50, 100%). VEPs were progressively greater in magnitude the greater the contrast of the viewed stimulus (2-way repeated measures ANOVA, n = 4 mice, effect of contrast, $F_{(4,12)}$ = 25.908, P < 0.001) both before and during PV+ neuronal activation using the hM4D(Gi) system (SNK post hoc test, 5 vs. 100 percent contrast, baseline; $q_{(4)}$ = 5.178, P = 0.013; CNO; $q_{(4)}$ = 17.595, P < 0.001, *Figure 4A*). Preservation of the approximately linear relationship of contrast and response during PV+ neuron inactivation indicates that responses have not exceeded their dynamic range.

We then induced SRP to different orientations in a different set of mice, one stimulus at 50% contrast and the other stimulus at 100% contrast. After 6 days of SRP at 50%, there was a modest but significant difference in VEP magnitude for familiar (188.81 ± 11.80 μV) and novel orientations (145.06 ± 8.11 μV, 2-way repeated measures ANOVA, n = 8 mice, SNK post hoc test, $q_{(7)}$ = 3.608, P = 0.023, *Figure 4B*), just as there was for the familiar (259.81 ± 17.66 μV) and novel orientations (192.81 ± 17.27 μV, SNK post hoc test, $q_{(7)}$ = 3.793, P = 0.018, *Figure 4C*) at 100% contrast. SRP expression was abolished by PV+ neuronal inactivation using the hM4D(Gi) system at 50% contrast as VEPs elicited by familiar (404.69 ± 59.17 μV) and novel orientations (376.94 ± 56.79 μV) were no longer significantly different (SNK post hoc test, $q_{(7)}$ = 2.289, P = 0.128, *Figure 4B*). The same was true at 100% contrast as VEPs evoked by the familiar (532.44 ± 63.82 μV) and novel orientations (514.88 ± 40.38 μV) were also no longer significantly different (SNK post hoc test, $q_{(7)}$ = 0.994, P = 0.494, *Figure 4C*). Again, the blockade of SRP expression was also clearly observed in the familiar/novel ratio, which dropped significantly from (1.31 ± 0.05) to (1.08 ± 0.08) with CNO application at 50% contrast (student's paired two-tailed t-test, $t_{(7)}$ = 2.983, P = 0.02, *Figure 4D*) and from (1.40 ± 0.10) to (1.02 ± 0.05) with CNO application at 100% contrast (student's paired two-tailed t-test, $t_{(7)}$ = 2.955, P = 0.021, *Figure 4E*). Thus, SRP could still be abolished through selective loss of PV+ neuron activity even at reduced contrasts eliciting submaximal responses. The disruption of SRP expression by inactivation of PV+ neurons is not a trivial consequence of a "ceiling effect".

## Activation of PV+ neurons also disrupts expression of SRP

PV+ neurons have been implicated in the sharpening of orientation selectivity in V1 (*Runyan et al., 2010*; *Adesnik et al., 2012*; *Atallah et al., 2012*; *Lee et al., 2012*; *Wilson et al., 2012*). If orientation selectivity were completely abolished by inactivation of PV+ neurons then the loss of SRP expression could be attributed to a failure of orientation discrimination rather than familiarity. A previous experiment has indicated that activation of PV+ neurons in V1 actually mildly enhances orientation-selectivity and visual discrimination (*Lee et al., 2012*). Therefore, we used optogenetics to activate PV+ neurons while mice were presented with familiar and novel stimuli to test whether SRP expression would be enhanced, unaffected or disrupted. PV+ neurons in binocular V1 of PV-Cre mice expressed Channel-rhodopsin 2 (ChR2) as a result of infection with AAV5-EF1α-DIO-hChR2 (H134R)-eYFP. During the same surgery VEP recording electrodes were implanted and optic fibers chronically implanted to deliver light to the recording site (*Figure 5A*). After stable expression 4 weeks after infection, mice were habituated to head-fixation on each of 2 days before undergoing a standard SRP experiment over 6 days (*Figure 5B,C*). Mice showed a significant increase in the magnitude of the average VEP evoked by the familiar oriented grating stimulus over days (Friedman repeated measures ANOVA on ranks, n = 11 mice, $X^2_{(5)}$ = 24.818, P < 0.001, *Figure 5C*). This potentiation was evident by day 2 (237.55 ± 29.01 μV) in comparison with day 1 (182.59 ± 21.97 μV, SNK post hoc test, $q_{(10)}$ = 7.675, P < 0.05). On day 7, mice were presented with interleaved blocks of

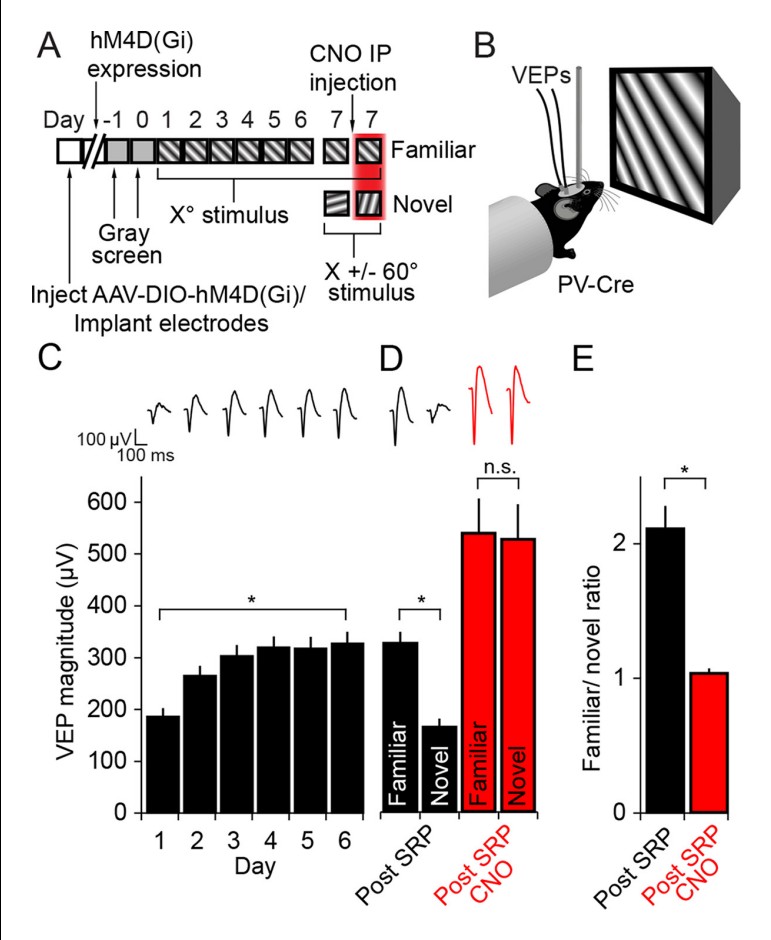

**Figure 3.** The expression of Stimulus-selective Response Potentiation (SRP) requires activity in PV+ neurons in V1. (**A**) Mice expressing hM4D(Gi) receptors in PV+ cells underwent a standard SRP induction protocol. On day 7, mice viewed a novel oriented stimulus in addition to the familiar stimulus; before and after CNO injection. (**B**) We acquired binocular VEPs from awake, head-fixed mice elicited by the same full-field oriented sinusoidal grating stimulus over several days. (**C**) As a result of multiple days of experience cortical response was dramatically potentiated such that the familiar stimulus evoked VEPs of significantly greater magnitude than the novel stimulus (black bars). (**D**) After application of CNO, VEPs underwent a notable increase in magnitude as a result of disinhibition (red bars). Most notably, CNO rendered response to familiar and novel stimuli equivalent in magnitude. (**E**) This lost discrimination of familiar and novel stimuli is reflected as a drop in the ratio of response to familiar and novel stimuli from approximately 2:1 (black) to approximately 1:1 (red). Significant comparisons are labeled with an asterisk and non-significant comparisons with n.s. throughout. Error bars are standard error of the mean (S.E.M.).

The following source data and figure supplements are available for figure 3:

**Source data 1.** PV Dreadds SRP induction and expression.
**Figure supplement 1.** CNO has no impact on SRP expression in WT mice.
**Figure supplement 1—source data 1.** CNO has no effect on SRP expression in WT mice.

familiar and novel stimuli. Also interleaved were blocks of familiar and novel stimuli in which blue light (473 nm) was continuously delivered via the optic fiber to the recording site in binocular V1. VEPs elicited by the novel X + 90° stimulus were significantly lower in magnitude (221.34 ± 33.55 µV) than those elicited by the familiar stimulus prior to optogenetic activation of PV+ neurons (310.70 ±

36.70 µV, 2-way repeated measures ANOVA, n = 11 mice, SNK post hoc test, $q_{(10)}$ = 11.799, P < 0.001, *Figure 5D*). Optogenetic activation of PV+ neurons significantly diminished the selectivity of SRP, as VEPs elicited by the familiar stimulus (181.23 ± 26.43 µV) and novel stimulus (157.80 ± 21.05 µV) were more similar in magnitude (interaction of stimulus x laser, $F_{(1,10)}$ = 46.606, P < 0.001; familiar vs. novel SNK post hoc test, $q_{(10)}$ = 3.094, P = 0.045, *Figure 5D*). The reduction of SRP selectivity is most clearly observed in the significant difference in the familiar/novel ratio without (1.55 ± 0.14) and with optogenetic stimulation (1.16 ± 0.07, student's paired two-tailed t-test, $t_{(10)}$ = 4.423, P = 0.001, *Figure 5E*). Laser stimulation did not affect the stimulus selectivity of SRP in wild type animals infected with virus (*Figure 5—figure supplement 1*), as there was no significant interaction between stimulus and laser (2-way repeated measures ANOVA, $F_{(1,8)}$ = 0.677, P = 0.434). These mice underwent SRP (*Figure 5—figure supplement 1A*) and on test day showed significantly larger VEPs to the familiar stimulus, both with the laser off (familiar VEP: 273.69 ± 28.25 µV, novel VEP: 200.19 ± 19.53 µV, 2-way repeated measures ANOVA, n = 9 mice, SNK post hoc test, $q_{(8)}$ = 5.234, P = 0.005) and the laser on (familiar VEP: 277.08 ± 29.80 µV, novel VEP: 194 ± 18.39 µV, $q_{(8)}$ = 5.916, P = 0.002, *Figure 5—figure supplement 1B*). Thus, SRP expression was disrupted with activation of PV+ neurons, just as it was with inactivation (*Figure 3D–E*). This result implies that PV+ neurons play a specific role in SRP expression beyond any role in enhancing orientation tuning.

## Full expression of SRP requires NMDA receptors (NMDARs) expressed in PV+ neurons

Multiple lines of evidence indicate that SRP is dependent upon the NMDA class of glutamate receptors (*Frenkel et al., 2006*; *Cooke et al., 2015*). Given the additional clear requirement for normal

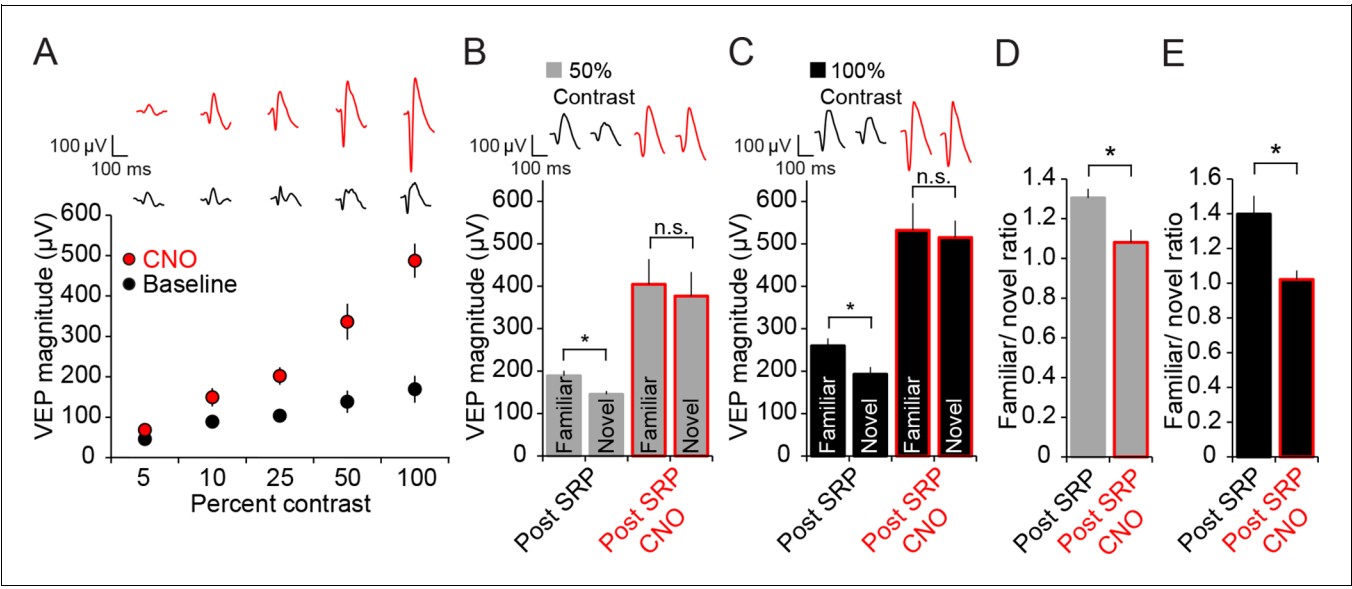

**Figure 4.** Expression of SRP to two separate contrast values is blocked by hM4D(Gi)-mediated PV+ neuron inactivation, but differential response to contrast is maintained. (**A**) CNO delivery to PV-Cre mice that had been infected with AAV9-hSyn-DIO-hM4D(Gi)-mCitrine impacted VEP magnitude (red) significantly across a range of contrast compared to baseline (black). (**B**) SRP was induced to two differently oriented stimuli, each at different contrast values: 50% (gray) and 100% (black). Modest but significant SRP was expressed at 50% contrast prior to CNO application. After CNO application (red outlines), VEPs increased significantly in magnitude but were no longer significantly different for familiar and a second novel orientation. (**C**) SRP was also expressed for a different orientation at 100% contrast and, again, VEP magnitude was increased and SRP blocked by delivery of CNO. (**D**) The blockade of SRP expression by CNO at 50% contrast was apparent in the reduction in the familiar/novel ratio for VEP magnitude. (**E**) The blockade of SRP expression at 100% contrast was also observed as a significant drop in familiar/novel ratio of VEP magnitude after CNO delivery. Significant comparisons are marked with an asterisk throughout while non-significant comparisons are marked with n.s. Error bars are standard error of the mean (S.E.M.).

The following source data is available for figure 4:

**Source data 1.** PV-neuron inactivation does not produce ceiling effect.

PV+ inhibitory cell function in SRP (*Figure 3D,E*) we selectively ablated the NMDAR from just PV+ cells by crossing the PV-Cre line of mice with a line in which the mandatory GluN1 subunit of NMDAR is excised by Cre recombinase activity (B6.129S4-*Grin1$^{tm2Stl}$*/J, GluN1 fl/fl). The progeny of this cross, in which both alleles of Grin1 (the gene encoding GluN1) were floxed, are henceforth described as either PV-GluN1 KO or Wildtype (WT)-GluN1 fl/fl depending on whether, respectively, Cre was expressed or not. We implanted PV-GluN1 KO (n = 14) and littermate WT-GluN1 fl/fl mice (n = 17) with VEP recording electrodes in layer 4, binocular V1. After recovery and 2 daily sessions of habituation we recorded VEPs elicited by an X° oriented grating stimulus. Immediately apparent was the significantly greater basal magnitude of VEPs recorded in the PV-GluN1 mice (242.45 ± 20.20 µV) relative to their littermate controls (130.88 ± 9.69 µV, student's two-tailed t-test, $t_{(29)}$ = -5.269, P < 0.001), consistent with the occurrence of disinhibition as a result of reduced glutamatergic drive on cortical PV+ inhibitory cells (*Figure 6A*). We then presented the same stimulus to these mice over several consecutive days and observed a significant difference in SRP across genotypes (2-way repeated measures ANOVA, interaction of genotype x day, $F_{(4,116)}$ = 3.835, P = 0.006, *Figure 6A*). Beyond day 3, VEP magnitudes were no longer significantly different between the PV-GluN1 KO mice (289.15 ± 24.57 µV) and WT-GluN1 fl/fl littermates (239.94 ± 20.72 µV, SNK post hoc test, $q_{(29)}$ = 2.242, P = 0.119), suggesting an occlusion of SRP by the already elevated basal VEP magnitude in the PV-GluN1 KO mice. The significant deficit in SRP is clearly apparent when the data are normalized to day 1 values (2-way repeated measures ANOVA, interaction of genotype x day, $F_{(4,116)}$ = 12.326, P < 0.001, *Figure 6B*). Again, a significant deficit in SRP emerged by day 3 in the PV-GluN1 KO mice (119.26 ± 10.13% day 1) compared with WT-GluN1 fl/fl littermates (183.33 ± 15.83% day 1, SNK post hoc test, $q_{(29)}$ = 4.727, P = 0.002), demonstrating that SRP is compromised by a loss of NMDAR expressed in PV+ neurons.

We also tested for the stimulus-selectivity of SRP expression by presenting both groups of animals with interleaved blocks of the familiar X° stimulus and a novel X + 90° stimulus (*Figure 6C*). Significant stimulus selectivity was present in both genotypes (2-way repeated measures ANOVA, stimulus, $F_{(1)}$ = 67.397, P < 0.001, interaction of genotype x stimulus, $F_{(1,29)}$ = 2.359, P = 0.135, *Figure 6C*): Although PV-GluN1 KO mice showed significant differences in VEP magnitude for familiar (340.24 ± 27.96 µV) and novel orientations (242.54 ± 24.40 µV, SNK post hoc test, $q_{(13)}$ = 6.372, P < 0.001) the difference was more pronounced in the WT-GluN1 fl/fl mice (SNK post hoc test, $q_{(16)}$ = 10.254, P < 0.001), in which the familiar stimulus elicited VEPs 318.74 ± 25.19 µV in magnitude and the novel stimulus elicited VEPs 176.06 ± 11.55 µV in magnitude. The familiar stimulus evoked VEPs that were not significantly different in the WT-GluN1 fl/fl mice (318.74 ± 25.19 µV) and the PV-GluN1 KO mice (340.24 ± 27.96 µV, SNK post hoc test, $q_{(29)}$ = 0.947, P = 0.507) but those evoked in the WT-GluN1 fl/fl by the novel stimulus (176.06 ± 11.55 µV) were significantly lower in magnitude than in the PV-GluN1 KO mice (242.54 ± 24.40 µV, SNK post hoc test, $q_{(29)}$ = 2.926, P = 0.045). The significant deficit in SRP expression in the PV-GluN1 KO mice is most apparent when the familiar/novel ratio of VEP magnitude (1.48 ± 0.13) is compared with the WT-GluN1 fl/fl littermates (1.83 ± 0.13, Mann Whitney rank sum test, U = 60.000, P = 0.020, *Figure 6D*). Thus, loss of NMDAR function selectively within PV+ neurons impairs the full expression of SRP.

## Adult OD plasticity does not require NMDARs expressed in PV+ neurons

We also examined if loss of NMDAR from PV+ neurons has an effect on OD plasticity. We implanted VEP recording electrodes in a second cohort of 11 PV–GluN1 KO and 7 WT-GluN1 fl/fl mice. After recovery and habituation over 2 days, monocular VEPs were acquired through each eye. After 7 days of MD, PV-GluN1 KO mice exhibited a significant potentiation of response through the open ipsilateral eye (187.91 ± 25.52 µV) compared with day 0 (90.64 ± 10.69 µV, 2-way repeated measures ANOVA, SNK post hoc test, $q_{(16)}$ = 8.849, P < 0.001, *Figure 6E*). Similarly, WT-GluN1 fl/fl mice also showed a significant potentiation of the open ipsilateral eye-response (154.57 ± 7.60 µV) after 7 days of MD compared with responses measured on day 0 (97.57 ± 15.85 µV, SNK post hoc test, $q_{(16)}$ = 4.137, P = 0.01, *Figure 6F*). Comparisons of OD ratios prior to MD in the adult PV-GluN1 KO (3.40 ± 0.29) and WT-GluN1 fl/fl mice (2.57 ± 0.33) did not reveal any significant difference (2-way repeated measures ANOVA, effect of genotype, $F_{(1)}$ = 3.424, P = 0.083). Significant shifts in the OD ratio occurred as a result of 7 days of MD in the adult PV-GluN1 KO (1.28 ± 0.33, SNK post hoc test, $q_{(16)}$ = 10.6, P < 0.001) and WT-GluN1 fl/fl mice (1.19 ± 0.09, SNK post hoc test, $q_{(16)}$ = 5.517, P =

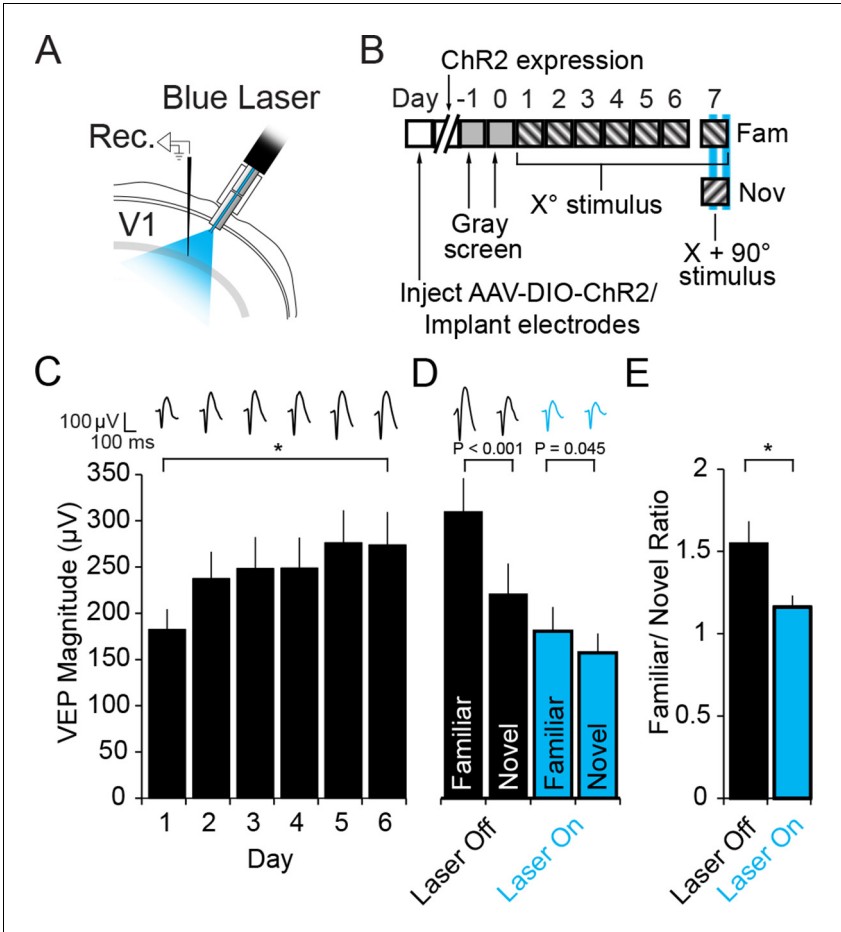

**Figure 5.** Optogenetic stimulation of PV+ inhibitory neurons prevents SRP expression. (**A**) Blue light was delivered locally into V1 via optic fibers chronically implanted at a 45° angle to target the VEP recording site in layer 4 of binocular V1 of PV-Cre mice infected with AAV5-EF1α-DIO-hChR2(H134R)-eYFP. (**B**) Experimental timeline showing that after viral infection, electrode implantation, and ChR2 expression; mice were accustomed to head-fixation and gray screen viewing. Subsequently, they underwent a standard SRP induction protocol over 6 days. On day 7, mice viewed a novel oriented stimulus in addition to the familiar stimulus and, on 50% of presentations of each stimulus, blue light (473 nm) was delivered to cortex to optogenetically activate PV+ cells. (**C**) Significant SRP was induced over 6 days as VEPs underwent a typical potentiation. (**D**) On day 7, SRP was expressed through significantly larger VEP magnitude in response to the familiar X° orientation than a novel X + 90° stimulus when blue light was not delivered (Black bars). In the presence of blue light (blue bars), VEPs were suppressed, and there was a significant reduction in the differential magnitude of VEPs driven by familiar and novel stimuli. (**E**) The ratio of VEP magnitude elicited by familiar/novel stimuli was significantly reduced by optogenetic activation of PV+ neurons, reflecting a decrement in SRP expression. Significant comparisons are marked with an asterisk and post hoc test p values are reported in **D** to emphasize the impact of laser stimulation on SRP selectivity. Error bars are standard error of the mean (S.E.M.).

The following source data and figure supplements are available for figure 5:

**Source data 1.** PV-neuronal activation.
**Figure supplement 1.** Blue light has no impact on SRP expression.
**Figure supplement 1—source data 1.** Laser does not effect VEPs in WT animals.

0.001). The shifted OD ratio also did not differ significantly across genotype (2-way repeated measures ANOVA, interaction of genotype and MD, $F_{(1,16)}$ = 2.638, P = 0.124, *Figure 6G*). Thus, the loss of NMDAR function from PV+ neurons did not have a significant impact on either the induction or the expression of adult OD plasticity, in contrast to SRP.

In order to confirm the removal of NMDARs selectively from PV+ cells, the PV-GluN1 KO mice were injected with an AAV5 vector to express GFP in a Cre-dependent fashion in PV+ cells only. After 1 month, fluorescence-guided intracellular recordings were performed from *ex vivo* slices of visual cortex. NMDAR-mediated synaptic transmission was normal in excitatory control cells but abolished in PV+ neurons (*Figure 6H*). This is expressed as significantly reduced NMDAR EPSC/AMPAR EPSC ratio in PV+ cells (0.040 ± 0.011, n = 8 cells from 5 mice) compared to neighboring non-fluorescent excitatory control cells (0.345 ± 0.039, n = 10 cells from 4 mice, student's one-tailed t-test, $t_{(16)}$ = -6.819, P < 0.001).

## Acute ketamine treatment reversibly eliminates SRP expression

A group of non-competitive, open-channel NMDAR blockers, including ketamine, PCP and MK801 are known to have the paradoxical impact of increasing net neuronal activity in the brain. It is thought that this apparent disinhibition arises from the preferential impact of these molecules on fast-spiking neurons, due to the tonic activation and the increased open-time of NMDAR expressed within these cells (*Homayoun and Moghaddam, 2007*; *Seamans, 2008*). Interestingly, these compounds are also psychotomimetic and can reproduce, at high sub-anesthetic doses, most of the symptoms of schizophrenia (*Krystal et al., 1994*). Here we tested the possibility that a single acute dose of one of these substances, ketamine, would have an impact on the expression of SRP due to its action on NMDAR expressed in PV+ fast spiking inhibitory neurons.

We implanted VEP recording electrodes in layer 4 of binocular V1 in a group of 10 C57BL/6 mice. After recovery and a standard SRP protocol we recorded VEP magnitudes driven by familiar and novel stimuli before, during and 2 days after recovery from systemic injection (i.p.) of a high but sub-anesthetic dose of ketamine (50 mg/kg) (*Figure 7A*). Ketamine had a significant effect on SRP expression (2-way repeated measures ANOVA, interaction of treatment x stimulus, $F_{(2,18)}$ = 29.479, P < 0.001, *Figure 7B*). After SRP but prior to ketamine delivery the familiar X° stimulus elicited VEPs of significantly greater magnitude (245.47 ± 21.00 μV) than a novel X + 60° stimulus (138.86 ± 9.32 μV, SNK post hoc test, $q_{(9)}$ = 9.228, P < 0.001) in these mice. After an hour of respite from head-fixation, mice were injected with ketamine and, 15 min later, they were returned to head-fixation and VEP magnitudes were again recorded. Under the influence of ketamine, the familiar X° stimulus elicited VEPs of significantly increased magnitude (381.90 ± 37.25 μV) relative to pre ketamine (245.47 ± 21.00 μV, SNK post hoc test, $q_{(9)}$ = 7.223, P < 0.001). However, VEPs elicited by a second novel X - 60° stimulus were increased relative to pre-ketamine by an even greater extent (397.53 ± 37.62 μV) relative to pre ketamine (138.86 ± 9.32 μV, SNK post hoc test, $q_{(9)}$ = 13.695, P < 0.001), such that VEPs were no longer significantly different in magnitude in response to familiar and novel stimuli (SNK post hoc test, $q_{(9)}$ = 1.353, P = 0.349). After 2 day's rest, allowing for complete recovery from drug effects, the mice were returned to the head-fixation apparatus and exposed to the familiar X° stimulus and a third novel stimulus, X + 90°. Just as observed prior to the ketamine injection, the X° stimulus evoked VEPs of significantly greater magnitude (267.46 ± 26.13 μV) than the novel X + 90° stimulus (149.40 ± 9.85 μV, $q_{(9)}$ = 10.219, P < 0.001) (*Figure 7A and B*).

This significant effect of ketamine on the discrimination of familiar and novel stimuli is summarized by the ratio of VEP magnitude driven by the familiar/novel stimulus (1-way repeated measures ANOVA, $F_{(2,18)}$ = 24.683, P < 0.001, *Figure 7C*). This familiar/novel ratio dropped significantly from 1.76 ± 0.10 prior to ketamine to 0.96 ± 0.02 after ketamine (SNK post hoc test, $q_{(9)}$ = 8.443, P < 0.001). Upon recovery, the familiar/novel ratio significantly recovered (1.79 ± 0.16, SNK post hoc test, $q_{(9)}$ = 8.759, P < 0.001) and was not significantly different from the first test after SRP but prior to ketamine injection (SNK post hoc test, $q_{(9)}$ = 0.316, P = 0.826). Thus, the psychotomimetic non-competitive NMDAR antagonist ketamine disrupts SRP. The fact that this is an acute effect on already established SRP that recovers after drug washout indicates that the role for NMDAR in PV+ fast-spiking GABAergic neurons may be in memory retrieval rather than learning.

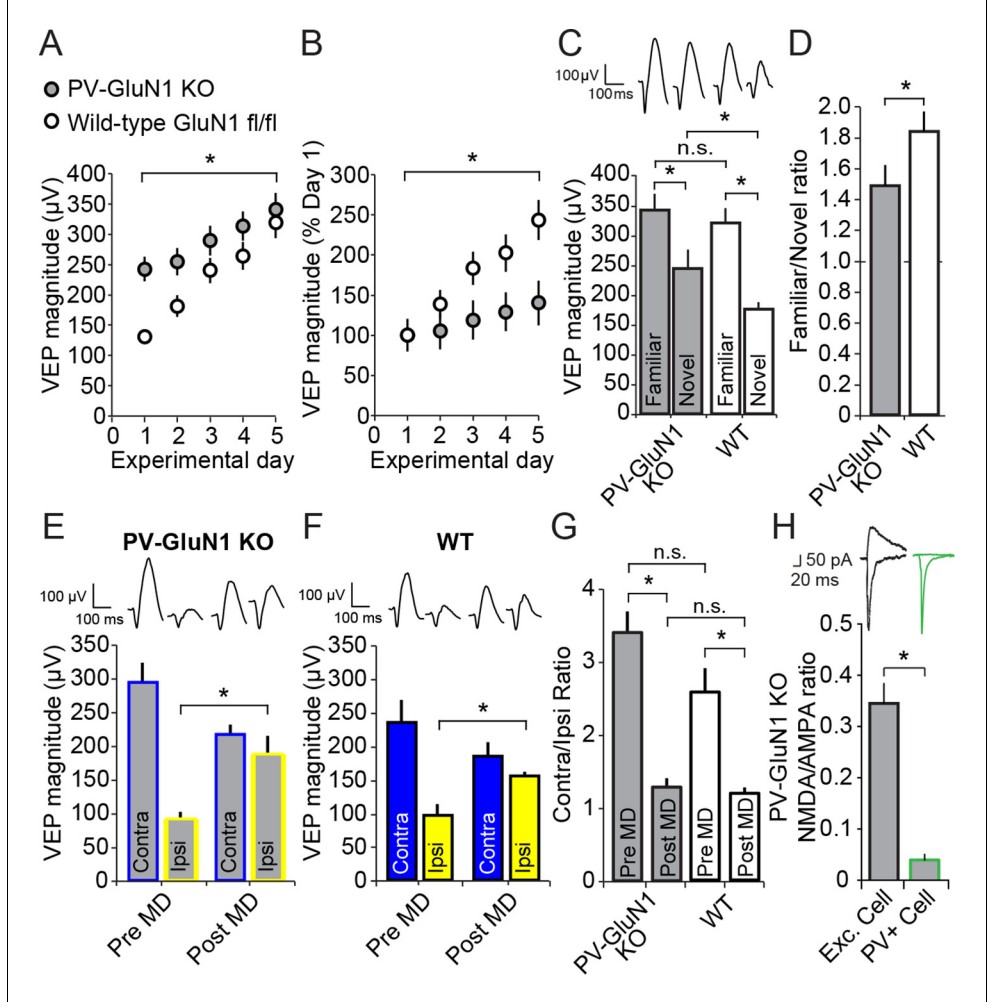

**Figure 6.** Loss of NMDA receptors selectively from parvalbumin+ cells impacts SRP but not adult OD plasticity. (**A**) VEPs recorded from mice in which the mandatory GluN1 subunit of the NMDA receptor was genetically ablated from PV+ cells using Cre recombinase technology (PV GluN1 KO, gray) were significantly greater in magnitude than those recorded from WT littermates (black), suggesting disinhibition of the visual response. In these same PV GluN1 KO mice, SRP was also significantly impacted as there is was significantly less gain in magnitude over days of repeated presentation of an X° stimulus than observed in WT littermates. (**B**) This significant reduction in the magnitude of SRP was most clearly observed if VEP magnitude was normalized to the magnitude on day 1. (**C**) After SRP, both PV GluN1 KO mice and their WT littermates exhibited a significantly greater VEP magnitude elicited by the now familiar stimulus than interleaved presentations of a novel oriented stimulus. However, consistent with the observed difference in magnitude on day 1, VEPs elicited by a novel X + 90° stimulus in PV GluN1 KO mice were significantly greater in magnitude than those in WT littermate mice. No significant difference was observed for VEPs elicited by the familiar stimulus. (**D**) A significant difference in the ratio of VEP magnitude elicited by the familiar and novel stimuli reveals a deficit in SRP expression in PV GluN1 KO mice. (**E**) In contrast, PV GluN1 KO mice exhibited a normal adult OD shift after 7 days of MD, resulting from open eye potentiation (yellow outlines). (**F**) Wild-type (WT) littermates exhibited the same significant open eye potentiation (yellow bars) after 7 days of MD. (**G**) A comparison of the degree of OD shift as a result of 7 days of MD reveals a significant shift in OD ratio in both genotypes but no difference between genotypes, indicating that NMDA receptors in PV+ cells are not required for induction or expression of the OD shift. (**H**) To confirm genetic ablation of NMDARs selectively from parvalbumin (PV+) neurons in the PV-GluN1 KO; mice were injected with an AAV5 vector to express GFP in a Cre-dependent fashion in PV+ cells only. After 1 month, fluorescence-guided intracellular recordings were performed from *ex vivo* slices of visual cortex. NMDAR-mediated synaptic transmission was normal in excitatory control cells but abolished in PV+ cells. This is expressed here as the NMDAR EPSC/AMPAR EPSC ratio in excitatory cells (black outline) and PV+ cells (green outline). Sample EPSC traces mediated by the AMPAR (downward) and NMDAR (upward) are shown at the top of the panel. Significant comparisons are marked with an asterisk throughout while non-significant comparisons are marked with n.s. Error bars are standard error of the mean (S.E.M.).

The following source data is available for figure 6:

**Source data 1.** PV-GluN1 KO

## Ketamine affects V1 responses through NMDARs expressed in PV+ neurons

Ketamine blocks NMDAR expressed in all cell types throughout the CNS and is also known to have targets other than the NMDAR (*Chen et al., 2009*). In order to determine if ketamine has its effect on V1 responses specifically through NMDAR expressed in PV+ cells, we tested if there was a differential effect of ketamine on VEP magnitude in PV-GluN1 KO mice and WT-GluN1 fl/fl mice. After a typical implantation and habituation protocol (*Figure 7D*) we tested VEP magnitudes elicited by a novel oriented X° stimulus in WT-GluN1 fl/fl mice (178.11 ± 38.41 µV, n = 8) and PV-GluN1 KO mice (260.79 ± 50.75 µV, n = 8) (*Figure 7E*). We then removed mice from head-fixation and allowed them to recover in their home-cage before delivering 50 mg/kg ketamine (i.p.). After 15 min, the mice were returned to head-fixation and we observed the impact of ketamine on VEPs elicited by a novel X + 90° oriented stimulus, which was significantly different in its effect on the two genotypes (2-way repeated measures ANOVA, interaction of genotype x treatment, $F_{(1,14)}$ = 18.454, P < 0.001). In the control WT-GluN1 fl/fl mice there was a significant potentiation of VEP magnitude as a result of ketamine application (386.65 ± 70.75 µV) in comparison to pre treatment (178.11 ± 38.41 µV, SNK post hoc test, $q_{(7)}$ = 8.505, P < 0.001, *Figure 7E*), replicating our previous finding (*Figure 7B*). However, in the PV-GluN1 KO mice, ketamine had no ostensible significant impact (258.67 ± 44.86 µV) in comparison to pre treatment (260.79 ± 50.75 µV, SNK post hoc test, $q_{(7)}$ = 0.087, P = 0.952). A ratio of VEP magnitudes pre and post ketamine treatment reveals the significant difference between ketamine's action on WT-GluN1 fl/fl mice (2.40 ± 0.25) and PV-GluN1 KO mice (1.12 ± 0.13, student's two-tailed t-test, $t_{(14)}$ = 4.610, P < 0.001) (*Figure 7F*). Thus, while ketamine has a wide range of effects in the CNS, it exerts itself on the response of V1 to visual input selectively through NMDAR expressed in PV+ cells.

## Ketamine has no effect on the expression of the adult OD shift

We then tested whether or not ketamine would disrupt either the OD ratio or the shift induced by 7 days of MD in the adult mouse. Again, C57BL/6 mice (n = 8) were implanted with VEP electrodes and taken through a standard surgery recovery and habituation protocol before measuring the OD ratio prior to 50 mg/kg ketamine and 15 min after ketamine delivery (*Figure 7G and H*). The normal contra/ipsi OD ratio exhibiting contralateral eye dominance prior to ketamine (2.84 ± 0.18) was not significantly altered by ketamine (2.61 ± 0.19, student's paired two-tailed t-test, $t_{(7)}$ = 0.871, P = 0.413) (*Figure 7I*). A separate group of mice (n = 11) then underwent a standard 7 day MD protocol. VEPs elicited by an X° oriented stimulus were recorded at baseline, followed by the deprivation period. After eye opening VEP magnitudes were re-tested with a novel X + 60°stimulus (the standard protocol to avoid contamination of the OD shift by SRP [*Frenkel and Bear, 2004*]). These mice were then tested again with a second novel X - 60°stimulus after 1 hr rest and an additional 15 min after systemic 50 mg/kg ketamine administration (*Figure 7J*). As expected, VEPs elicited through the open ipsilateral eye (93.73 ± 12.85 µV) were significantly potentiated by 7 days of MD in the adult mouse (155.73 ± 15.86 µV, 2-way repeated measures ANOVA, SNK post hoc test, n = 11 mice, $q_{(10)}$ = 7.102, P < 0.001), reflecting the well-documented OD shift. These same ipsilateral VEPs were then further potentiated by ketamine administration (232.03 ± 22.19 µV, SNK post hoc test, $q_{(10)}$ = 8.741, P < 0.001). However, ketamine had a similar significant potentiating effect on the contralateral VEP (279.07 ± 14.53 µV), relative to pre-ketamine (180.14 ± 11.22 µV, SNK post hoc test, $q_{(10)}$ = 11.333, P < 0.001, *Figure 7K*), suggesting a uniform scaling of response through the two eyes. This observation is confirmed by the fact that the OD ratio, significantly shifted from (2.84 ± 0.29) to (1.23 ± 0.11, Friedman repeated measures ANOVA on ranks, n = 11 mice, $X^2_{(2)}$ = 16.545, p<0.001, SNK post hoc test, $q_{(10)}$ = 5.126, P < 0.05) by 7 days of MD, was not further significantly affected by delivery of ketamine (1.28 ± 0.09, SNK post hoc test, $q_{(10)}$ = 0.426, P > 0.05, *Figure 7L*). Thus, while ketamine prevents SRP expression through action on NMDAR expressed in PV+ cells, it does not significantly affect the adult OD shift after 7 days of MD, consistent once again with PV+ cells contributing to the expression of SRP but not adult OD plasticity.

## Visual novelty detection requires PV+ neuron activity within V1

Given the clear involvement of PV+ neurons in the expression of SRP we wanted to determine if loss of PV+ neuronal function local to V1 would have any behavioral impact. Head-restrained mice

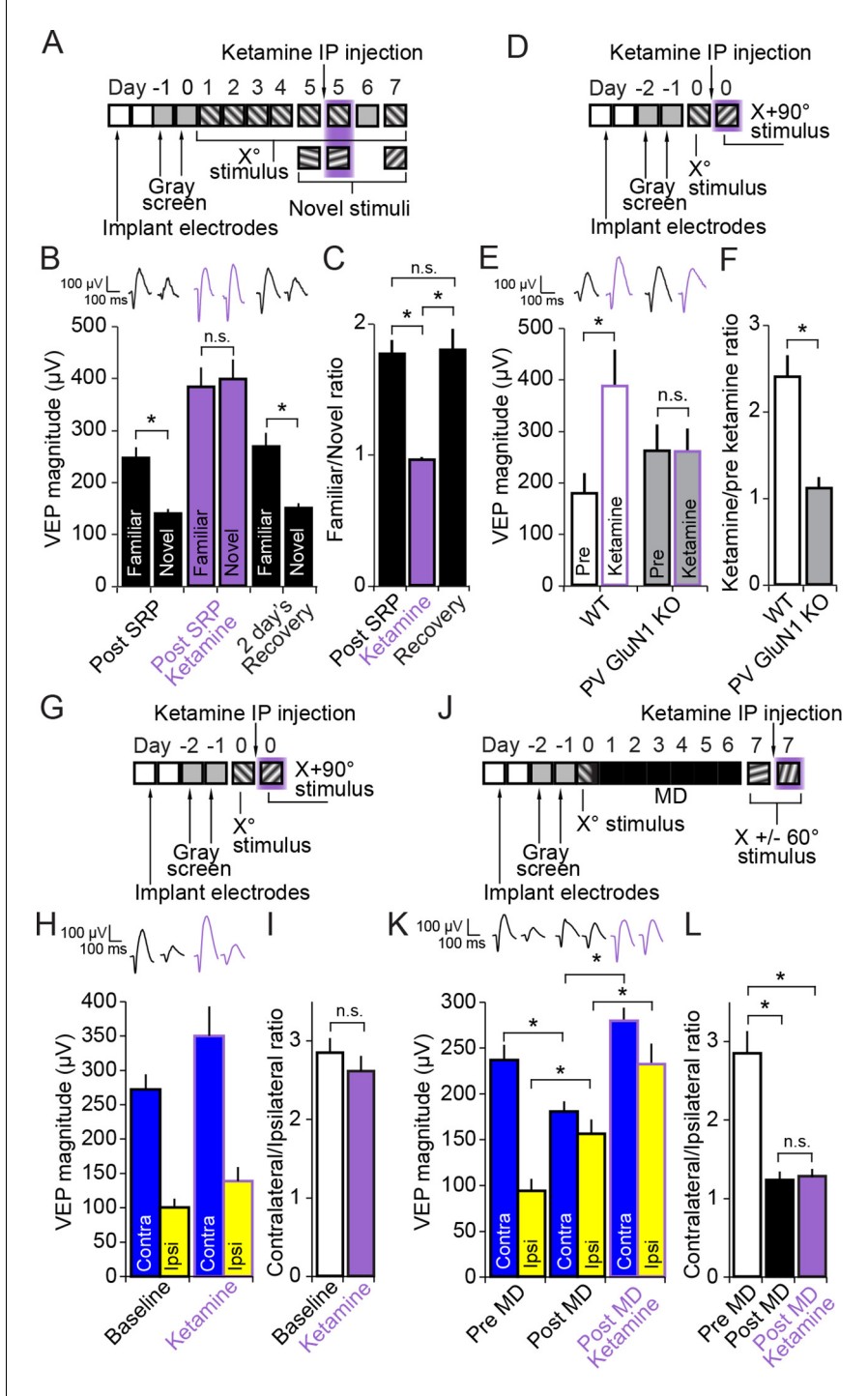

**Figure 7.** Ketamine prevents expression of SRP through blockade of NMDA receptors expressed in parvalbumin+ neurons but does not impact expression of the adult OD shift. (**A**) Mice were bilaterally implanted with VEP recording electrodes in layer 4 of binocular V1. After habituation to head-fixation and a gray screen for 2 days, SRP was induced over 4 days by repeatedly presenting sessions of an X° stimulus. On day 5, SRP expression was tested by presenting interleaved blocks of the familiar X° stimulus and a novel X + 60° stimulus. In order to test the acute impact of blocking NMDA receptors on SRP expression, 50 mg/kg of ketamine was then delivered systemically 15 min before re-acquiring VEPs elicited by the familiar X° stimulus and interleaved presentations of a second novel X - 60° stimulus. Mice were then allowed 2 days recovery and a complete washout of ketamine before re-testing SRP expression by again testing VEP magnitude in response to the familiar X° stimulus and a third novel X + 90°

*Figure 7 continued on next page*

*Figure 7 continued*

stimulus on day 7. (**B**) Significant SRP was expressed on experimental day 5 prior to ketamine delivery as the familiar X° stimulus elicited VEPs of greater magnitude than the novel stimulus. Delivery of 50 mg/kg ketamine (purple) had two notable impacts on the VEP: First, the overall magnitude of the VEP increased. Second, and most importantly, the significant difference in magnitude of VEPs elicited by familiar and novel stimuli was no longer present. This effect was acute, as SRP expression was again significantly apparent 2 days later. (**C**) The ratio of VEP magnitude elicited by the familiar stimulus over the novel. This ratio was close to 2 and not significantly different prior to or after recovery from ketamine administration but dropped significantly to approximately 1 during ketamine exposure. (**D**) We tested whether ketamine had a differential impact on VEP magnitude in PV GluN1 KO mice and WT littermate mice. (**E**) In the WT littermate mice (white bars) 50 mg/kg ketamine had a significant potentiating effect on VEP magnitude, consistent with our previous observation. In contrast, ketamine had no significant impact on VEP magnitude in the PV GluN1 KO mice (gray bars). (**F**) The selectivity of ketamine's impact on the WT mice is observed by comparing the ratio of VEP magnitude during ketamine over baseline, which was significantly greater for WT mice than the PV GluN1 KO mice, in which the ratio was approximately 1. (**G**) In a separate group of mice, a similar protocol was then used to determine whether the OD ratio is affected by ketamine. (**H**) Ketamine impacted both the VEPs driven through the contralateral eye (blue) and ipsilateral eye (yellow) equally. (**I**) This scaled effect is demonstrated by a lack of significant difference between OD ratios prior to (white) and during 50 mg/kg ketamine (purple). (**J**) We next tested whether ketamine has any impact on the expression of adult OD plasticity by recording VEP magnitudes through either eye in a new group of adult mice before taking them through a standard 7 day MD protocol. (**K**) As anticipated, 7 days of contralateral eye MD induced a significant ipsilateral eye potentiation (yellow) and ketamine then further potentiated VEPs elicited through both contralateral (blue) and ipsilateral eyes. (**L**) The OD ratio shifts significantly from a ratio heavily biased towards the contralateral eye, to a less biased ratio. Ketamine administration did not significantly affect the magnitude of the OD shift. Significant comparisons are marked with an asterisk throughout while non-significant comparisons are marked with n.s. Error bars are standard error of the mean (S.E.M.).

The following source data is available for figure 7:

**Source data 1.** Ketamine impact on cortical plasticity.

---

viewing a phase reversing visual grating stimulus are known to exhibit a stereotyped motor response called a visually-induced fidget, or vidget (*Cooke et al., 2015*). Vidgets can be measured via a piezo-electric sensor located beneath the forepaws of the mouse (*Figure 8A*). Importantly, it has been shown that the magnitude of the vidget response is inversely correlated to the familiarity of the visual stimulus. That is, a visual stimulus that is very familiar to the animal will on average evoke a relatively weak vidget behavioral response. In contrast, the presentation of a novel stimulus will on average evoke a vidget of significantly greater magnitude. Therefore, this behavioral response reflects the ability of the animal to discriminate and respond to a novel visual stimulus in its environment. Importantly, genetic and pharmacological manipulations local to V1 that inhibit SRP also disrupt the behavioral discrimination of familiar and novel stimuli (*Cooke et al., 2015*), demonstrating that this differential vidget response to familiar and novel stimuli is dependent on the plasticity in visual cortex. Since PV+ neuron inactivation disrupted the expression of SRP, we tested whether this manipulation would likewise disrupt visual novelty detection.

A group of PV-Cre mice expressing hM4D(Gi) receptors in PV+ cells (n = 19), underwent a SRP protocol similar to the previous experiment (*Figure 3A*) in which mice viewed phase-reversing gratings of a particular orientation (X° stimulus) each day for 6 days. On the 7th day mice viewed blocks of the now familiar visual stimulus interleaved with blocks of a novel oriented stimulus (X + 60°). On day 7, vidget behavioral responses were acquired via a piezoelectric device situated underneath the forepaws of the mice (*Figure 8A*), in order to measure the animal's discrimination of familiar and novel stimuli. On day 8, PV+ cells in V1 were then inactivated by systemic delivery of CNO (i.p.) and vidget responses were acquired to the familiar X° stimulus and a second X - 60° novel stimulus. Inactivation of PV+ neurons in V1 significantly affected stimulus discrimination (2-way repeated measures ANOVA, interaction of treatment x stimulus, $F_{(1,18)} = 13.644$, P = 0.002, *Figure 8B*): Prior to CNO, mice exhibited significantly larger vidget responses to the novel visual stimulus (3.64 ± 0.32 a.u.) than the familiar stimulus (1.95 ± 0.26 a.u., SNK post-hoc test, $q_{(18)} = 9.237$, P < 0.001). During PV+ cell inactivation in V1, behavioral responses to the familiar (2.01 ± 0.16 a.u.) and novel visual stimuli (2.46 ± 0.22 a.u.) were no longer significantly different (SNK post-hoc test, $q_{(18)} = 2.503$, P = 0.086).

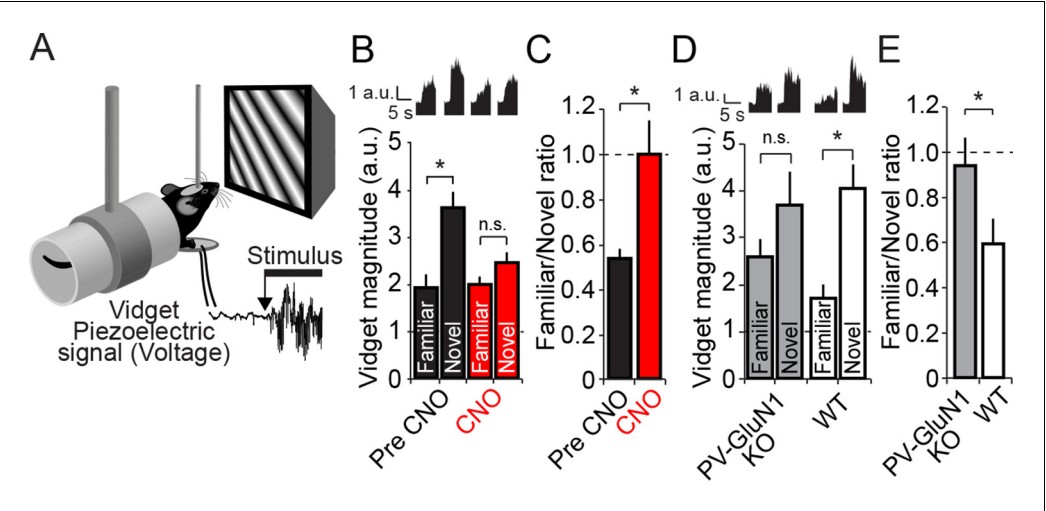

**Figure 8.** Discrimination of familiar and novel oriented stimuli involves PV+ neurons in V1 and NMDAR expressed within PV+ neurons. (**A**) Using the same protocol as described in *Figure 3B*, mice were progressively familiarized with a specific oriented stimulus. On the test day, as mice viewed familiar and novel stimuli, vidget behavioral responses were measured via a piezoelectric sensor located beneath the forepaws of the head-fixed mouse. (**B**) After a standard SRP protocol, mice expressing hM4D(Gi) receptors selectively within PV+ cells of binocular V1 were exposed to both familiar and novel stimuli. Prior to application of CNO, mice exhibited significant behavioral evidence of discriminating this familiar orientation from interleaved presentations of a novel oriented stimulus (black bars). After application of CNO, there was no longer successful discrimination of familiar and novel stimuli (red bars). Averaged vidget responses are displayed at the top of this panel. (**C**) A significant difference was observed in the ratio of response to familiar and novel stimuli from pre CNO (black) to post CNO (red). (**D**) A deficit in OSH was also apparent in PV GluN1 KO mice as vidget recordings demonstrated a failure to significantly discriminate familiar from novel orientations (gray bars). WT littermates exhibited significantly greater vidget magnitudes for novel than familiar stimuli, indicating unimpaired discrimination of familiarity from novelty (white bars). Averaged behavioral responses are displayed above with accompanying scale bars. (**E**) The significant deficit of PV GluN1 KO mice in discriminating familiar from novel stimuli is apparent in the ratio of behavior elicited by the familiar over the novel stimulus in comparison to WT littermates. Significant comparisons are marked with an asterisk throughout while non-significant comparisons are marked with n.s. Error bars are standard error of the mean (S.E.M.).

The following source data and figure supplements are available for figure 8:

**Source data 1.** Novelty detection after PV+ neuronal disruption.
**Figure supplement 1.** Cumulative distributions of average vidget behavioral response to familiar and novel stimuli for each individual animal included in analysis presented in *Figure 8D* and *Figure 8E*.
**Figure supplement 1—source data 1.** Per animal plots of Novelty detection after PV+ neuronal disruption.

The deleterious effect of PV+ cell inactivation in V1 on the animal's ability to discriminate familiar and novel stimuli is most obvious when a ratio of response to familiar/novel visual stimuli is calculated. Prior to CNO delivery this ratio was significantly lower (0.54 ± 0.04) than during CNO treatment (1.00 ± 0.15, Wilcoxon signed rank test, W = 138.000, Z = 2.777, P = 0.004, *Figure 8C*). These findings indicate that PV+ neuron activity is required not only for the expression of SRP, but also for visual novelty detection.

## Discrimination of familiar and novel stimuli requires NMDAR within PV+ neurons

Finally, given the observation that the discrimination of familiar and novel visual stimuli is diminished by inactivation of PV+ neurons in V1, we tested whether a similar failure would occur in the PV-GluN1 KO mice (n = 17) compared with WT-GluN1 fl/fl littermates (n = 15, 2-way repeated measures

ANOVA, stimulus, $F_{(1)} = 14.632$, $P < 0.001$, *Figure 8D*). In the PV-GluN1 KO mice, the familiar stimulus produced less behavioral response ($2.75 \pm 0.42$ a.u.) than a novel stimulus ($3.67 \pm 0.79$ a.u.). However, the difference was not significant (SNK post hoc test, $q_{(16)} = 2.570$, $P = 0.079$). In comparison, the concurrently tested WT-GluN1 fl/fl littermate mice showed a suppression of behavioral response to the familiar stimulus ($1.76 \pm 0.26$ a.u.) relative to the novel stimulus ($4.08 \pm 0.51$ a.u.) that was significant (SNK post hoc test, $q_{(14)} = 5.008$, $P = 0.001$). This observation was reinforced by a comparison of the ratio of behavior produced by familiar and novel stimuli (*Figure 8E*), in which there was a significant difference between the PV-GluN1 KO mice ($0.59 \pm 0.11$) and their WT-GluN1 fl/fl littermates ($0.94 \pm 0.14$, student's one-tailed t-test, $t_{(30)} = -1.992$, $P = 0.028$), reflective of the deficit in discrimination of familiar and novel stimuli as a result of lost NMDAR function in PV+ cells. Individual animals' average vidget responses to familiar and novel stimuli (*Figure 8—figure supplement 1*) reveal significantly decreased stimulus selectivity subsequent to CNO administration (*Figure 8—figure supplement 1A–B*), and in the PV GluN1 KO mice as compared to WT-GluN1 fl/fl littermate controls (*Figure 8—figure supplement 1C–D*). Thus, not only do NMDAR in PV+ cells contribute to SRP expression but they are also involved in its behavioral correlate.

## Discussion

Our experiments reveal a surprising role for PV+ inhibitory neuron activity in the expression of SRP but not of eye dominance or deprivation-enabled potentiation of the non-deprived eye after MD in the adult mouse. The data show that NMDARs in PV+ neurons are critically important for the expression of SRP, and that familiarity recognition and novelty detection measured behaviorally are compromised both by loss of PV+ neuron activity within V1 and loss of NMDAR from PV+ neurons.

### PV+ interneurons and OD plasticity in V1

There has been a long-standing interest in the possible roles of PV+ neurons in juvenile OD plasticity. These include modulation of the sensitivity of visual cortex to the effects of MD (*Fagiolini et al., 1994*; *Hanover et al., 1999*; *Huang et al., 1999*; *Chattopadhyaya et al., 2004*) and the functional expression of the OD shift (*Maffei et al., 2006*; *Yazaki-Sugiyama et al., 2009*; *Smith and Bear, 2010*). Our findings indicate that expression of OD plasticity in adult mice does not require participation of the PV+ neurons. In interpreting the current findings, it is important to recognize how juvenile and adult OD plasticity differ. In juvenile mice (< ~P35), a rapid and reliable consequence of MD is depression in cortex of responses mediated by the deprived eye. This is followed by a progressive compensatory increase in responses through the non-deprived eye (*Frenkel and Bear, 2004*). In adult mice maintained under standard laboratory conditions, depression of deprived-eye responses can be a weak and variable consequence of MD, but potentiation of the non-deprived eye still occurs reliably (*Sawtell et al., 2003*; *Sato and Stryker, 2008*). This deprivation-enabled response potentiation is not mediated by homeostatic synaptic scaling because it requires visual experience through the non-deprived eye (*Blais et al., 2008*), activation of cortical NMDA receptors (*Sawtell et al., 2003*; *Sato and Stryker, 2008*), and is unaffected (in adults) by genetic deletion of TNFα that abolishes scaling (*Ranson et al., 2012*). These features are consistent with an alternative hypothesis that the adult OD shift occurs because deprivation of the dominant contralateral eye causes a metaplastic shift in the LTP threshold, enabling visual experience through the weaker ipsilateral eye to drive "Hebbian" synaptic strengthening (*Cooper and Bear, 2012*). This interpretation is supported by evidence that light deprivation promotes LTP in visual cortex (*Kirkwood et al., 1996*; *Philpot et al., 2001*; *2003*) and that αCaMKII mutants that lack LTP also lack adult OD plasticity (*Ranson et al., 2012*), and it is compatible with our finding that expression of the adult OD shift is not dependent on the activity of PV+ inhibitory neurons.

Saiepour and colleagues also recently observed that ocular dominance index (ODI) values of V1 neurons are not altered by PV+ interneuron suppression in monocularly deprived adult mice (*Saiepour et al., 2015*). Using single unit recordings they first showed that monocular deprivation caused a shift in ODI values towards the non-deprived eye, as expected. They then tested the dependence of this shift on PV+ cell activity by measuring ODI values of V1 neurons during optogenetic suppression of PV+ cells. This optogenetic-suppression did not alter the deprivation-induced shift in ODI values. Our results using VEP recordings and PV+ suppression via hM4Di Dreadd receptors are in agreement with these results from Saiepour and colleagues, demonstrating that

suppression of PV+ cell activity after deprivation does not interfere with the expression of the adult OD shift. Although the findings that PV+ neurons are not necessary for expression of OD plasticity do not rule out a modulatory role in the induction mechanisms, we note that the adult OD shift still occurred in mutant mice in which PV+ neurons lack NMDARs.

## PV+ interneurons and SRP in V1

In contrast to the adult OD shift, we found that the expression of SRP relies on PV+ neuron circuitry. This finding came as a surprise, as SRP reflects the potentiation of short-latency synaptic currents and increased spiking activity in layer 4 (*Cooke et al., 2015*), and previous experiments had strongly indicated that SRP might be considered a naturally occurring form of LTP. For example, induction of SRP is input-specific, dependent upon NMDA receptor activation and α-amino-3-hydroxy-5-methyl-4-isoxazolepropionic acid receptor (AMPAR) insertion in principal cells (*Frenkel et al., 2006*), and reversed by infusion of the zeta-inhibitory peptide (ZIP) that also erases LTP (*Cooke and Bear, 2010*). Furthermore, LTP elicited by theta-burst stimulation of the thalamus occludes and is occluded by SRP (*Cooke and Bear, 2010*). Altogether, these findings suggest a modification of feed-forward excitatory synaptic transmission.

Before assigning a specific role to inhibition in the expression of SRP, two relatively trivial explanations must be addressed. The first is that the loss of discrimination of familiar and novel orientations could be accounted for simply by a failure of orientation selectivity and not by a failure of memory. Throughout the experiments presented here we have used orientations that are at least 60° and often 90° different, meaning that orientation discrimination should be as unchallenging as possible. Given the modest impact on orientation tuning of suppressing PV+ interneuron activity in V1 (*Runyan et al., 2010*; *Adesnik et al., 2012*; *Wilson et al., 2012*) we believe that this explanation is improbable. However, we took a direct approach to testing this possibility by stimulating activity in PV+ cells using channelrhodopsin-2 (ChR2) while mice are viewing familiar and novel oriented stimuli. Stimulation of PV+ inhibition has been shown to actually mildly enhance orientation-selectivity (*Lee et al., 2012*) and yet our experiments reveal that the difference between visual cortical response to familiar and novel orientations is significantly diminished by activation PV+ neurons. This result casts PV+ neurons in a very specific role in the recognition of familiar stimuli in addition to modest participation in orientation-selectivity.

A second concern is that, after the suppression of PV+ inhibition, V1 may be operating outside the dynamic range where discrimination is possible. However, we show that the cortex responds to input from the contralateral eye with a response of approximately double the magnitude of that elicited by input through the ipsilateral eye even after silencing the PV+ cells, suggesting that the response of the cortex is not saturated. We also find that PV+ neuron silencing affects SRP even when it is induced using lower contrast stimuli that evoke submaximal responses. Moreover, it is also worth noting that less specific pharmacological approaches to suppressing inhibition in V1 (*Khibnik et al., 2010*) result in much greater magnitude of response to visual input than we observe here for PV+ neuron inactivation, PV-GluN1 KO or ketamine application, indicating that our manipulations do not encroach on a physiological ceiling.

One relatively simple explanation for the current findings is that SRP results from lasting homosynaptic (*i.e.* input-specific) long-term depression (LTD) of glutamatergic input to PV+ inhibitory neurons. Thereafter, presentation of a familiar stimulus evokes a greater response in V1 due to selective disinhibition of cortical circuitry. This hypothesis is consistent with our observation that SRP is deficient if NMDARs are no longer expressed on PV+ cells. However, because we used a non-reversible genetic modification in our initial PV-GluN1 KO experiment, we were initially unable to show if these receptors were important for the storage of information or the retrieval of information already stored. Ketamine, which among other actions serves as a non-competitive NMDAR antagonist, has been shown to preferentially antagonize NMDARs on PV+ cells (*Homayoun and Moghaddam, 2007*; *Seamans, 2008*). By using ketamine to induce a temporary blockade of these receptors, which we confirmed by showing that there was no impact of ketamine on V1 responses in the PV-GluN1 KO mice, we were able to demonstrate that interference with NMDARs on PV+ cells disrupts *expression* of SRP even after it has been induced and saturated. Importantly, acute application of ketamine did not affect the maintenance or stability of stored information, because a strong bias for the familiar stimulus returned after ketamine washout. If SRP is indeed accounted for by homosynaptic depression of excitatory synapses carrying information about the familiar stimulus orientation, this

LTD would need to be expressed by a synapse-specific reduction in NMDAR-mediated responses in order to account for these findings.

However, this simple model is difficult to reconcile with the findings reviewed above that SRP depends on LTP-like mechanisms, *e.g.*, NMDAR-dependent AMPAR delivery in principal cells. Considered together, the findings to date indicate that induction and maintenance of SRP occur by modification of excitatory synapses onto excitatory neurons, but that expression of SRP is mediated by stimulus-selective modulation of PV-cell activity in V1. Obviously a polysynaptic circuit is required to yield these properties, and dissecting this essential cortical circuitry will require much more work. However, it is interesting to note a recent study by Makino and Komiyama who showed, in addition to a stimulus-selective reduction in PV+ neuron activity in V1 as mice were repeatedly exposed to a specific visual stimulus, an experience-dependent increase in the responsiveness of somatostatin-positive interneurons (*Makino and Komiyama, 2015*). This result is particularly interesting as it is known that many interneuron types innervate one another and, therefore, that the activity of these cell types are interdependent. It is not difficult to sketch circuits in which stimulus-selective, feed-forward polysynaptic inhibition of PV+ cells accounts for SRP expression. Clearly, experiments aimed at teasing apart the involvement of other interneuron types in SRP should now be a high priority.

We note that other studies have suggested that PV+ neurons may be important for plasticity following repeated exposure to sensory stimuli. For instance, Meyer and colleagues, studying infero-temporal cortex (IT) in monkeys, showed that familiarity resulted in a sharpening of cortical responses to dynamic visual stimuli, which was likely orchestrated by local fast-spiking putative PV+ cells (*Meyer et al., 2014*). Additionally, mice constitutively lacking neuronal activity-regulated pentraxin (NARP), a synaptic protein specifically enriched at excitatory synapses on PV+ cells, displayed a reduction in excitatory synapses onto PV+ cells and impairment in SRP-like effects (*Chang et al., 2010*; *Gu et al., 2013*). Intriguingly, however, the NARP KO mice also failed to exhibit both juvenile and adult OD plasticity. In juvenile animals, it is well established that reducing inhibition can impair induction of OD plasticity (*Ramoa et al., 1988*) and delay the developmental decline in the mechanism of deprived-eye depression (*Hanover et al., 1999*; *Huang et al., 1999*). We speculate that the loss of OD plasticity in the adult NARP KO may be related to disruption of the developmental switch from juvenile to adult-type OD plasticity (*Heimel et al., 2011*).

Our findings indicate that although open-eye potentiation as a result of MD in the adult and SRP are superficially similar, the expression mechanisms of these two forms of plasticity are quite distinct. Based on evidence provided here as well as by others (*Sawtell et al., 2003*; *Ranson et al., 2012*; *Saiepour et al., 2015*), the deprivation enabled potentiation of the open eye is consistent with a potentiation of synapses between excitatory cells and is independent of PV+ cell inhibition. However, further work will be necessary to confirm that open eye potentiation in the adult animal is expressed by synaptic alterations between excitatory cells, and to determine whether these processes are present at the level of thalamocortical synapses or intracortical synapses.

## PV+ interneurons and familiarity recognition

SRP correlates with OSH (*Cooke et al., 2015*), a form of visual learning which is dependent upon NMDARs in V1, and which enables the animal to make the critically important discrimination between familiarity and novelty. We have shown here that modulation of PV+ neuron activity in V1 is required for this selective recognition of familiarity and consequent novelty detection. Loss of NMDAR function in these same cells also impairs familiarity recognition. The cognitive symptoms of schizophrenia, and other psychiatric disorders, are characterized by deficits in habituation and familiarity (*Braff et al., 1995*; *Ramaswami, 2014*) and individuals with schizophrenia display impairment in visual cortical plasticity (*Çavuş et al., 2012*). Dysfunction of PV+ inhibitory neurons and the NMDARs expressed on PV+ cells have also been implicated in these same psychiatric disorders through a range of approaches (*Lewis et al., 2005*; *Gogolla et al., 2009*; *Lewis et al., 2011*). This notably includes observations of the profound psychotomimetic effect in humans of non-competitive NMDAR antagonists such as ketamine (*Krystal et al., 1994*; *Krystal, 2015*), a substance that, as we show here, prevents SRP expression. We suggest that understanding the cortical physiology underlying the processes of familiarity recognition and novelty detection may yield great insight into dysfunction underlying some symptoms of these disorders and that, because these processes are as important to mice as they are to humans, they are likely to be experimentally tractable.

# Materials and methods

## Mice

All procedures adhered to the guidelines of the National Institutes of Health and were approved by the Committee on Animal Care at MIT, Cambridge, MA, USA. For all experiments mice were male, aged between P60-90 and on a C57BL/6 background (Charles River laboratory international, Wilmington, MA). They were housed in groups of 2–5 with food and water available *ad libitum* and maintained on a 12 hr light-dark cycle. For hM4D(Gi) experiments, mice were Parvalbumin-Cre recombinase knock-in mice (B6;129P2-*Pvalb$^{tm1(cre)Arbr}$*/J, PV-Cre) on a C57BL/6 background. For GluN1 knockdown experiments, these PV-Cre mice were bred with homozygous mice in which the Grin1 gene, which encodes the GluN1 sub-unit was flanked by LoxP sites (B6.129S4-*Grin1$^{tm2Stl}$*/J, GluN1 fl/fl), enabling Cre-dependent ablation of this mandatory subunit of the NMDA receptor. Multiple generations were required to set up crosses yielding offspring that were homozygous GluN1 fl/fl and Cre-expressing. Of these approximately 50% expressed Cre and 50% served as wild-type littermates on the GluN1 fl/fl background. All experiments were conducted blind to genotype and treatment.

## Surgery

Mice were first injected with 0.1 mg/kg Buprenex sub-cutaneously (s.c.) to provide analgesia. They were then anesthetized with an intraperitoneal (i.p.) injection of 50 mg/kg ketamine and 10 mg/kg xylazine. Prior to surgical incision, 1% lidocaine hydrochloride anesthetic was injected under the mouse's scalp. The skull was then cleaned with iodine and 70% ethanol. A steel head post was affixed to the skull anterior to bregma using cyanoacrylate glue. Burr holes (< 0.5 mm) were then drilled in the skull over binocular V1 (3.1 mm lateral of lambda). Tapered tungsten recording electrodes (FHC, Bowdoinham, ME, US), 75 μm in diameter at their widest point, were implanted in each hemisphere, 450 μm below cortical surface. Silver wire (A-M systems, Sequim, WA, US) reference electrodes were placed over prefrontal cortex. Mice were allowed to recover for at least 24 hr prior to head-fixation.

## Viral infections

For hM4D(Gi) experiments, we infected V1 of P30-60 PV-Cre mice or wild-type littermates with an AAV9-hSyn-DIO-HA-hM4D(Gi)-IRES-mCitrine virus (UNC viral core – generated by Dr. Brian Roth's laboratory) and for optogenetic experiments we infected mice from the same line with AAV5-EF1α-DIO-hChR2(H134R)-eYFP (UNC viral core – generated by Dr. Karl Deisseroth's laboratory). Using a glass pipette and nanoject system (Drummond scientific, Broomall, PA, US), we delivered 81 nl of virus at each of 3 cortical depths: 600, 450, and 300 μm from the cortical surface, and allowed 5 min between re-positioning for depth. Mice were allowed 3–4 weeks recovery for virus expression to peak before experiments were initiated.

## Drug delivery

Clozapine-N-oxide (CNO, Enzo Life Sciences) was diluted in saline and injected i.p. at a dosage of 5 mg/kg 30 min prior to stimulus delivery. Ketamine hydrochloride (Vedco) was diluted in water and delivered i.p. at 50 mg/kg 15 min prior to stimulus delivery.

## Stimulus delivery

Visual stimuli consisted of full-field, 100% contrast, sinusoidal gratings that were presented on a computer monitor. Visual stimuli were generated using custom software (http://bearlab-s1.mit.edu/supp6/PlxStimOne_Opto.zip) written in either C++ for interaction with a VSG2/2 card (Cambridge Research systems, Kent, U.K.) or Matlab (MathWorks, Natick, MA, U.S.) using the PsychToolbox extension (http://psychtoolbox.org) to control stimulus drawing and timing. The display was positioned 20 cm in front of the mouse and centered, thereby occupying 92° × 66° of the visual field. Visual stimuli consisted of full-field sinusoidal grating stimuli phase reversing at a frequency of 2 Hz. Mean stimulus luminance was 27 cd/m$^2$. Grating stimuli spanned the full range of monitor display values between black and white, with gamma-correction to ensure constant total luminance in both gray-screen and patterned stimulus conditions. For data described in *Figures 1–3* and

*6*, experiments were fully automated and each stimulus block consisted of 200 phase reversals with 30-s intervals between each stimulus presentation, during which the screen was gray but of equivalent luminance. For experiments described in *Figures 4* and *5* stimulus blocks were also 200 phase reversals in length but each was triggered directly by the experimenter, meaning that the interval was approximately 30-s. Throughout, stimulus orientation varied such that a novel orientation was always a minimum of 25° different from any experienced previously by the individual mouse and was never within 20° of horizontal because these orientations are known to elicit VEPs of greater magnitude than vertical or oblique stimuli. If more than 1 orientation was shown within a session, stimuli were pseudo-randomly interleaved such that 3 consecutive presentations of the same stimulus never occurred. For monocular presentation an occluding paddle was positioned in front of one eye to limit stimulus presentation to the opposite eye.

## In vivo electrophysiology

VEP recordings were conducted in awake, head-restrained mice. Prior to recording, mice were habituated to the restraint apparatus by sitting in situ in front of a gray screen for a 30-minute session on each of two consecutive days. For experiments in which monocular VEPs were subsequently acquired, mice were also habituated to the occluding paddle positioned in front of each eye. On stimulus presentation days, mice were presented with 5 blocks of 200 phase reversals of each oriented stimulus separated by gray screen presentation for ~30 s. For monocular presentations, recordings were conducted in sequence for each eye. VEP magnitude was then quantified by measuring trough-peak response magnitude averaged over all phase reversals.

Recordings presented in *Figures 1–3* and *6* were amplified and digitized using the Recorder-64 system (Plexon Inc., Dallas, TX). Two recording channels were dedicated to recording EEG/VEPs from V1 in each implanted hemisphere and a third recording channel was reserved for the Piezo-electrical input carrying the behavioral information. Recordings presented in *Figures 4* and *5* were amplified using DAM80 amplifiers (World Precision Instruments, Florida, U.S.) and digitized using a custom National Instruments system (National Instruments, Texas, U.S.)). Local field potential was recorded from V1 with 1 kHz sampling and a 500 Hz low-pass filter. Data was extracted from the binary storage files and analyzed using custom software (http://bearlab-s1.mit.edu/supp6/VEP_Analysis.zip) written in C++, Matlab and Labview. VEPs were averaged across all phase reversals within a block and trough-peak difference measured during a 200-millisecond period from phase reversal.

## Ex vivo electrophysiology

For experiments concerning Dreadd receptor expression in PV+ cells presented in *Figure 1*: Three weeks after infection with AAV9-hSyn-DIO- HA-hM4D(Gi)-IRES-mCitrine virus, visual cortical slices were prepared from the injected mice as previously described (*Philpot et al., 2001*). After dissection 350-μm-thick coronal slices recovered for 30 min at 32°C and then for an additional 2 hr at room temperature, in a holding chamber filled with warmed artificial cerebrospinal fluid (ACSF), which contained: 124 mM NaCl, 3.5 mM KCl, 1.25 mM $Na_2PO_4$, 26 mM $NaHCO_3$, 1.2 mM $MgCl_2$, 2 mM $CaCl_2$, and 10 mM dextrose, saturated with 95% $O_2$, 5% $CO_2$. Whole cell patch clamp recordings were performed at 30°C from the parvalbumin-positive interneurons in the layer 4, identified by the ECFP fluorescence. Pipette tip resistances were 3–5 MΩ. Internal solution contained: 20 mM KCl, 100 mM Na-gluconate, 10 Hepes, 4 mM MgATP, 0.3 mM Na2GTP, 7 mM phosphocreatine-Tris, 0.2% biocytin with pH adjusted to 7.2 and osmolarity adjusted to 300 mOsm. All recordings were made using Axopatch 200B (Molecular Devices, Sunnyvale, CA) at 10 kHz sampling rate. Cells with the series resistance <30 MΩ were included for analysis. Data analysis was done using pClamp (Molecular Devices) and custom-written python scripts. For experiments concerning NMDAR EPSCs and AMPAR EPSCs in PV-GluN1 KO mice as presented in *Figure 6H*: 4 PV-GluN1 KO mice were infected with AAV5-EF1α-DIO-hChR2(H134R)-eYFP, to allow for fluorescence-guided intracellular recordings of PV+ cells. Four weeks after infection, visual cortical slices were prepared as describe above. For fluorescently labeled PV+ cells and unlabeled neighboring excitatory cells: EPSCs were recorded at -70 mV and +40 mV with pipettes filled with balanced intracellular solutions (in mM): 115 cesium methane-sulfonate (CsMeSO3), 2.8 NaCl, 0.4 EGTA, 4 ATP-$Mg^{2+}$, 10 $Na^+$-phosphocreatine, 0.5 $Na^+$-GTP, 5 TEA-$Cl^-$, 5 QX-314 $Br^-$ buffered with 20 HEPES, pH 7.25, osmolarity 290 mOsm. Synaptic events were evoked by stimulation of the white mater (150 μs, 0.1 Hz, glass electrode, and

stimulator A365 from WPI). AMPA component was measured from EPSCs at -70 mV in the presence of PTX (100 μM) and glycine (1 μM) and NMDA component was measured from EPSCs at +40 mV in the presence of DNQX (20 μM).

## Behavior

All behavioral experiments were performed during the mouse subject's light cycle. A piezo-electrical recording device (C.B. Gitty, Barrington, NH, USA) was placed under the forepaws of head-restrained mice during all sessions. Mice became accustomed to the apparatus by sitting in situ in front of a gray screen for a 30-min session on each of 2 days. Before stimulus presentation on each day mice also underwent 5 min of gray screen presentation after the experimenter had left the room. A continuous voltage signal was recorded from the piezo device for the entire session. Movements were detected as a shift in the voltage signal. The recording system was automated so that no one was ever present in the closed room for any of the recording period and white noise was played at 67 dB in order to mask outside noise.

For vidget scoring, the continuous voltage signal was down-sampled to 100 Hz. The period of interest in the experiments described here lasted from 2 s prior to stimulus onset until 5 s after stimulus onset (which was the first 10 phase reversals in a block). Quantification of movement driven by the onset of the stimulus (the vidget) was calculated by taking the Root Mean Square ($SQRT(X^2)$) of the voltage signal. Post-stimulus signal was then normalized to the average magnitude during the 2-s period prior to stimulus onset. The average normalized magnitude across the 5-s period subsequent to stimulus presentation was then used to quantify the degree of stimulus-driven movement and this is described throughout in arbitrary units (a.u.).

## Optogenetics

After viral infection mice were also bilaterally implanted with VEP recording electrodes positioned in layer 4. Ready-made optic fibres (200 μm girth) mounted in stainless steel ferrules (1.25 mm diameter, 2 mm fibre projection, Thor labs, Newton, NJ, US) were then implanted positioned lateral (3.5 mm lateral to lambda) to the recording site and at a 45° angle to the recording electrode, 0.1 mm below surface in each hemisphere. 1 month later, after peak of viral expression, mice were habituated to the head-fixation apparatus over 2 days before conducting optogenetic experiments. We delivered 31-s long continuous pulses of blue light (473 nm, 150 mW) into V1 using a laser (Optoengine LLC, Midvale, UT, US). These light pulses were delivered simultaneous to 50% of the 30-s long visual stimulus presentations, commencing 30 ms prior to visual stimulus onset and ending 30 ms after offset. Animals were sacrificed and perfused within a week after this experiment for histological analysis.

## Immunohistochemistry

Mice were deeply anaesthetized with fatal plus (pentobarbital) and perfused with saline followed by 4% paraformaldehyde in 0.1 M phosphate buffer. The brain was removed and post-fixed for 24 hr at room temperature. After fixation, the brain was sectioned into 60 μm coronal slices using a vibratome. Slices were incubated with blocking solution (10% fetal bovine serum in PBS with 0.2% Triton X-100) for 1 hr at room temperature and then with anti-Parvalbumin mouse primary antibody (MAB1572, Millipore, Billerica, MA; 1:1000) and anti-GFP antibody (Ab290, Abcam, Cambridge, United Kingdom; 1:7000) details diluted in blocking solution overnight at 4 degrees Celsius. Slices were then washed three times with PBS and incubated with the secondary antibody for 1 hr at room temperature (Alexa488-conjugated anti-rabbit IgG, Invitrogen, Carlsbad, CA; 1:500, Alexa594-conjugated anti-mouse IgG, Invitrogen; 1:500). Slices were washed three times with PBS and mounted with 49,6-diamidino-2 phenylindole (DAPI)-containing Vectashield (Vector Laboratories, Burlingame, CA). Fluorescence images were taken with a confocal fluorescence microscope (Olympus, Tokyo, Japan).

## Statistics

In the results section, all data is expressed as a mean ± standard error of the mean (S.E.M). Sigma-plot was used for statistical analysis. Normality of distribution and homogeneity of variation was tested and parametric ANOVAs (for multiple groups) or t-tests (for 2 groups) were performed when

data passed these tests. Otherwise, non-parametric ANOVAs or t-tests on ranks were used. If ANOVAs yielded significance, Student-Newman-Keuls post-hoc tests with adjustment for multiple comparisons were applied for individual comparisons. Repeated measures ANOVAs or paired t-tests were applied for all within subject comparisons. For other comparisons unpaired ANOVAs or t-tests were used. Individual tests used are described in the results. $P < 0.05$ is used as a threshold for significance throughout but exact P values are given for all comparisons for which the P value is above 0.001. Post-hoc power analyses were used to determine that comparisons reached a power of $> 0.8$ for an alpha value of 0.05. Technical replicates were only used as N for the ex vivo electrophysiological test of hM4D(Gi) efficacy, when 10 cells were recorded from 7 mice, and the test of NMDAR functional loss in PV-GluN1 KO mice, when 8 cells were recorded from 5 mice. Throughout the remainder of the study we report the N is an individual animal (biological replicate). Technical replicates (secondary samples within each animal) were not undertaken for in vivo electrophysiology. Although both hemispheres were implanted initially, a decision was made as to the best hemisphere, based on response magnitude, morphology and binocularity after the very first recording (prior to any measure of plasticity). Technical replicates were undertaken for the behavior, with 10, 6 (for PV-HM4D(Gi) experiments) or 5 stimulus onsets (for PV-GluN1 KO experiment) being delivered per day, although we only report the daily average (biological replicate). There variations in protocol were due experiments being conducted by different experimenters at different times. However, protocols were completely consistent across treatments/genotypes. The individual technical replicates are available in the uploaded source data files.

## Acknowledgements

This research was supported by the Howard Hughes Medical Institute, the National Eye Institute (5R01EYO23037) and The Picower Institute Innovation Fund. ESK was supported in part by National Institute for Mental Health training grant (1T32MH074249), and RWK was supported in part by the Junior Faculty Development Program of the JPB Foundation. We thank Marie Carlen for helping to initiate our use of PV-GluN1 KO mice. We also thank Arnie Heynen, Jason Shepherd and Emily Osterweil for helpful scientific discussions. We express our ongoing gratitude to Erik Sklar, Amanda Coronado, and Gerald for technical assistance and Suzanne Meagher for invaluable administrative support.

## Additional information

### Funding

| Funder | Grant reference number | Author |
|---|---|---|
| National Eye Institute | 5R01EYO23037 | Eitan S Kaplan<br>Sam F Cooke<br>Robert W Komorowski<br>Alexander A Chubykin<br>Lena A Khibnik<br>Mark F Bear |
| National Institute of Mental Health | 1T32MH074249 | Eitan S Kaplan |
| Howard Hughes Medical Institute | | Eitan S Kaplan<br>Sam F Cooke<br>Robert W Komorowski<br>Alexander A Chubykin<br>Lena A Khibnik<br>Mark F Bear |

The funders had no role in study design, data collection and interpretation, or the decision to submit the work for publication.

### Author contributions

ESK, SFC, Conception and design, Acquisition of data, Analysis and interpretation of data, Drafting or revising the article; RWK, Conducted unit recordings presented in Figure 1, Analysis and

interpretation of data, Drafting or revising the article; AAC, Conducted intracellular recordings conducted in Figure 1, Acquisition of data, Analysis and interpretation of data, Drafting or revising the article; AT, Conducted intracellular recordings conducted in Figure 6, Acquisition of data, Analysis and interpretation of data, Drafting or revising the article; LAK, Conception and design, Drafting or revising the article, Contributed unpublished essential data or reagents; JPG, Provided critical expertise for the optogenetic experiments conducted in Figure 5, and provided methodological resources, Conception and design, Drafting or revising the article, Contributed unpublished essential data or reagents; MFB, Conception and design, Analysis and interpretation of data, Drafting or revising the article

## Author ORCIDs
Mark F Bear, http://orcid.org/0000-0002-9903-2541

## Ethics
Animal experimentation: This study was performed in strict accordance with the Guide for the Care and Use of Laboratory Animals of the National Institutes of Health. All procedures were approved by the institutional animal care and use committee (IACUC) protocol (#0612-050-15), and the Department of Comparative Medicine at the Massachusetts Institute of Technology. All surgery was performed under ketamine/xylazine anesthesia, and every effort was made to minimize suffering.

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
