## [Decision Letter]

Thank you for submitting your work entitled "Contrasting Roles for Parvalbumin-Expressing Inhibitory Neurons in Two Forms of Adult Visual Cortical Plasticity" for consideration by *eLife*. Your article has been reviewed by three peer reviewers, including Thomas Mrsic-Flogel, who is a member of our Board of Reviewing Editors, and overseen by Gary Westbrook as the Senior Editor.

The reviewers have discussed the reviews with one another and the Reviewing Editor has drafted this decision to help you prepare a revised submission.

Summary:

This manuscript provides evidence that PV-expressing interneurons do not contribute to the expression of an ocular dominance shift in adult V1, but are involved in the expression of selective response potentiation in V1 as shown with visual evoked responses in V1 and at the behavioral level. The authors also show that ketamine interferes with the expression of SRP, but not with the expression of an adult OD shift. Finally the authors provide evidence for the contribution of NMDA-receptors on PV interneurons to SRP and its prevention by ketamine. The experiments provide novel insights into how PV-expressing interneurons contribute to different forms of plasticity in adult V1. The study will appeal to scientists working on cortical plasticity and provides information relevant for understanding psychiatric disorders (especially schizophrenia).

Essential revisions:

Here is the summary of the main suggestions for revision, which we think you will be able to address in your revision.

1) The suggestion that ketamine acts mostly on the NMDAR receptors of PV interneurons is interesting (and potentially relevant to schizophrenia research). However, the claim that NMDAR in PV cells are solely responsible for the ketamine effect on SRP is not convincing, as it is possible that ketamine also affects SRP through NMDA receptors on excitatory neurons or by acting through sites other than the NMDA receptor. The reviewers therefore think it would be pertinent to investigate whether GluN1 was knocked out in all PV interneurons. If not, it would explain the leftover SRP in the PV GluN1 KO mice (and possibly the leftover effect of ketamine on SRP). One way (but not the only way) to test the completeness of KO in the floxed line is to use GluN2A immunocytochemistry. The logic here is that GluN2A remains cytoplasmic in NMDAR1 deficient cells. One could then in principle detect whether there is a fraction of PV cells that do not have GluN2A in the cytoplasm. This control (or some other control experiment addressing the cell-type specificity of ketamine dependence of SRF) might be sufficient to explain your results and strengthen your claims.

2) All reviewers found that the discussion about mechanisms underlying SRP was not sufficiently specific or detailed. For example, it is not clear where in the circuit the changes are occurring, and whether SRP relates to spiking or is only apparent at the synaptic level. In particular, do stimulus-selective changes in VEPs reflect enhanced spiking of principal neurons, of inhibitory neurons, changes of feed-forward thalamocortical synaptic drive onto which neuron class, changes of top-down modulation (e.g. Makino and Komiyama 2015)? It is also not clear whether PV cells are required for the induction of SRP, which would be predicted by some models of cortical plasticity. Finally, it is not sufficiently discussed how PV neuron activity could modulate VEPs in a stimulus-specific manner while accounting for the fact that the Bear lab has previously shown that LTP-like phenomena (AMPAR insertion, NMDAR-dependence, occlusion by thalamocortical LTP) are critical for SRP and OSH expression (Frenkel et al. 2006, Cooke and Bear 2010). We therefore strongly encourage you to propose a coherent, defendable and detailed mechanistic view about SRP, and how this is different from OD plasticity, in the Discussion.

3) Can the (variability of) behavior of the animal influence the measurement and quantification of SRP results? The reviewers request additional quantification to ensure that behavioral differences cannot explain neural response differences (i.e. does the VEP response precede any behavioral response?).

4) We encourage you to acknowledge the findings by Saiepour 2015 (expression of OD plasticity not strongly affected by PV manipulations). Although this work reduces some of the novelty of the results about OD plasticity, the reviewers still found it an important dataset in this manuscript.

---

## [Author Response]

*This manuscript provides evidence that PV-expressing interneurons do not contribute to the expression of an ocular dominance shift in adult V1, but are involved in the expression of selective response potentiation in V1 as shown with visual evoked responses in V1 and at the behavioral level. The authors also show that ketamine interferes with the expression of SRP, but not with the expression of an adult OD shift. Finally the authors provide evidence for the contribution of NMDA-receptors on PV interneurons to SRP and its prevention by ketamine. The experiments provide novel insights into how PV-expressing interneurons contribute to different forms of plasticity in adult V1. The study will appeal to scientists working on cortical plasticity and provides information relevant for understanding psychiatric disorders (especially schizophrenia).*

*Essential revisions:*

*Here is the summary of the main suggestions for revision, which we think you will be able to address in your revision. 1) The suggestion that ketamine acts mostly on the NMDAR receptors of PV interneurons is interesting (and potentially relevant to schizophrenia research). However, the claim that NMDAR in PV cells are solely responsible for the ketamine effect on SRP is not convincing, as it is possible that ketamine also affects SRP through NMDA receptors on excitatory neurons or by acting through sites other than the NMDA receptor. The reviewers therefore think it would be pertinent to investigate whether GluN1 was knocked out in all PV interneurons. If not, it would explain the leftover SRP in the PV GluN1 KO mice (and possibly the leftover effect of ketamine on SRP). One way (but not the only way) to test the completeness of KO in the floxed line is to use GluN2A immunocytochemistry. The logic here is that GluN2A remains cytoplasmic in NMDAR1 deficient cells. One could then in principle detect whether there is a fraction of PV cells that do not have GluN2A in the cytoplasm. This control (or some other control experiment addressing the cell-type specificity of ketamine dependence of SRF) might be sufficient to explain your results and strengthen your claims.*

We are aware that besides acting on NMDARs on PV+ cells, ketamine acts upon a variety of other molecular targets, including NMDARs expressed on principal cells. We thank the reviewers for highlighting our lack of clarity on this issue. The purpose of our application of ketamine in the PV-GluN1 knockout was to demonstrate that, in specific regard to increases in the magnitude of the VEP, the selective loss of NMDAR function in PV+ cells was sufficient to completely abolish the impact of ketamine, which is the result that we show in Figure 7 and F. A range of other physiological consequences of ketamine may well be unimpaired by loss of NMDAR function selectively in PV+ cells but are not the focus of investigation in this particular study. We note that the effect of ketamine to significantly increase VEP magnitude in wild-type mice is paradoxical because a glutamate receptor blocker might be expected to reduce excitatory transmission in the brain. This finding is, however, consistent with other observations that the drug reduces cortical inhibition and therefore increases cortical excitation by acting preferentially (but by no means exclusively) on GABAergic fast-spiking cells in the cortex through NMDAR (Houmayoun 2007; and reviewed by Seamans 2008).

The reviewers suggest that there may be some residual SRP in the PV-GluN1 KO mice due to incomplete genetic deletion of the receptors in those cells. In order to address this question directly, we conducted fluorescence-guided intracellular recordings from ex vivo slices if V1 to confirm the loss of NMDAR-mediated currents in PV+ cells from the PV-GluN1 KO mice. As expected, recordings from PV+ interneurons in V1 revealed a lack of any significant NMDAR-mediated excitatory postsynaptic currents (EPSCs), while showing normal AMPAR-mediated EPSCs (Figure 6). Conversely, neighboring principal cells displayed both NMDAR and AMPAR-mediated EPSCs. The selective loss of NMDAR EPSCs in PV+ neurons in the PV-GluN1 KO mice is reflected by plotting the NMDAR/AMPAR EPSC ratio in PV+ cells as well as neighboring principal cells. Thus, PV+ neurons sampled from V1 in the PV-GluN1 knockout mice failed to express NMDA receptors. This data has been added to the manuscript as Figure 6.

Unfortunately, immunohistochemistry for any of the NMDAR subunits proved extremely challenging (largely due to the quality of commercially available antibodies) and we were unable to convince ourselves that this method could be used to confirm that every single PV+ neuron in V1 lacked NMDAR function. However, given the consensus within the literature on ketamine’s actions through NMDAR expressed within PV+ neurons, we believe that the fact that we observed no effect whatsoever of ketamine on VEP magnitude in the PV-GluN1 KO mice, in contrast to their wild-type littermates, indicates that the genetic deletion of GluN1 within PV+ cells must be close to complete across all PV+ neurons within V1 of these knockout animals.

We agree that it is an important issue to understand more deeply what mechanism has accounted for the remaining SRP that exists in the PV-GluN1 knockout mouse, which is not observed with acute ketamine blockade. As a result of our confirmation that NMDAR-mediated currents are completely absent from at least all sampled PV+ cells in the PV-GluN1 KO mice, we hypothesize that the residual potentiation in the PV-GluN1 may derive from some form of developmental compensation that could occur through another subclass of neuron or another receptor type. Acute manipulation of NMDARs via ketamine would not allow for such compensation to occur, and therefore could explain the complete abolishment of SRP expression in the ketamine experiment that was not apparent in the PV-GluN1 knockout. It is possible that in SRP, NMDARs may be involved not only on PV+ cells but also on principal cells (Frenkel 2006). In this case, the effect of ketamine on NMDARs expressed on principal cells may also contribute to the complete lack of SRP expression in these experiments (even if it cannot account for the action of ketamine on VEP magnitude), in comparison to the PV-GluN1 KO mouse experiments. In order to test this hypothesis, future studies will probe the involvement of NMDARs on principal cells in various cortical layers in SRP. The alternative explanation, that another class of receptor compensates for the loss of NMDAR in the PV-GluN1 KO is also worthy of further investigation.

*2) All reviewers found that the discussion about mechanisms underlying SRP was not sufficiently specific or detailed. For example, it is not clear where in the circuit the changes are occurring, and whether SRP relates to spiking or is only apparent at the synaptic level. In particular, do stimulus-selective changes in VEPs reflect enhanced spiking of principal neurons, of inhibitory neurons, changes of feed-forward thalamocortical synaptic drive onto which neuron class, changes of top-down modulation (e.g. Makino and Komiyama 2015)? It is also not clear whether PV cells are required for the induction of SRP, which would be predicted by some models of cortical plasticity. Finally, it is not sufficiently discussed how PV neuron activity could modulate VEPs in a stimulus-specific manner while accounting for the fact that the Bear lab has previously shown that LTP-like phenomena (AMPAR insertion, NMDAR-dependence, occlusion by thalamocortical LTP) are critical for SRP and OSH expression (Frenkel et al. 2006, Cooke and Bear 2010). We therefore strongly encourage you to propose a coherent, defendable and detailed mechanistic view about SRP, and how this is different from OD plasticity, in the Discussion.*

We appreciate the need for clear overall insight and have modified the Discussion section to accommodate those possible interpretations of the underlying mechanisms of SRP that we can currently envisage. It is becoming increasingly clear that this is a far more complex phenomenon than we had previously realized and may, therefore, be more interesting in many ways.

Previously published data has provided evidence that SRP is apparent at the synaptic level as well as being expressed as changes in neural spiking. Multi-unit recordings from Layer 4 of wild-type animals showed that peak firing rate in response to a familiar stimulus was significantly elevated compared to those evoked by a novel stimulus (Cooke et al., 2015). In response to a phase reversing visual stimulus, the negative-going component of the VEP occurs simultaneously with a short latency spiking response to the visual stimulus. Familiar stimuli drive larger VEPs and increased neural spiking in Layer 4 in comparison to novel stimuli. By using laminar probes that span the cortical layers, we have also explored how familiarity changes the laminar flow of current sinks and sources and using current source density analysis (Cooke et al., 2015). From this we have observed alterations across multiple V1 layers resulting from experience, but the most striking effects are apparent in the short-latency layer 4 current sink, where familiar stimuli drive a larger current sink and increase the spiking of L4 neurons. Our current findings indicate that this increased spiking in L4 in part derives from removal of inhibition mediated by PV+ neurons.

Although it is known that experience drives changes in synaptic and spiking activity, we do not yet know how changes in spiking relate directly to individual cell types, which is an area of active investigation. Previously published data has also revealed that although there is evidence for thalamocortical (TC) LTP-like phenomena as a mechanism underlying SRP (Frenkel et al., 2006, Cooke and Bear 2010), selectivity for familiar and novel stimuli cannot be fully explained by changes occurring at the TC synapse in Layer 4. Isolation of the TC synaptic response by silencing intra-cortical spiking revealed a loss of SRP expression, suggesting that although changes may be occurring at the level of the TC synapse, downstream intra-cortical network alterations are involved (Cooke and Bear, 2014). We propose that these LTP-like changes previously reported are critical in the induction and maintenance of the stored information related to the familiar stimulus. Alterations in PV+ cell activity are necessary, however, for the retrieval/expression of this information. Since the major focus of this paper was on the expression mechanisms of SRP, the role of PV+ cells in the induction process remains unknown but we plan to pursue this further. A discussion of this hypothesis and how it differs from adult OD plasticity is presented in the revised Discussion section of the manuscript.

*3) Can the (variability of) behavior of the animal influence the measurement and quantification of SRP results? The reviewers request additional quantification to ensure that behavioral differences cannot explain neural response differences (i.e. does the VEP response precede any behavioral response?).*

The reviewers certainly raise an understandable concern, that differences in cortical physiology could be reflective of behavioral events rather than the other way round. In our previous paper (Cooke et al. 2015, Nature Neuroscience) we have discussed this issue at length and presented data showing that the visually induced behavioral response (vidget) commences ~150 ms after stimulus onset, a good 100 ms after the peak negativity of the first VEPs that we record. More importantly, we have reported that behavioral responses do not occur on all stimulus presentations and for the 5 stimulus onsets (gray to phase reversing grating) that occur on each recording day, only a subset will produce a behavioral response. However, the VEPs evoked during each of these stimulus blocks are similar in magnitude, regardless of whether a behavioral response has occurred or not and the difference in cortical response between familiar and novel stimuli is maintained even for all of those stimulus blocks where no behavior occurred. Thus, the behavior itself does not account in any way for VEP magnitude or the phenomenon of SRP.

*4) We encourage you to acknowledge the findings by Saiepour 2015 (expression of OD plasticity not strongly affected by PV manipulations). Although this work reduces some of the novelty of the results about OD plasticity, the reviewers still found it an important dataset in this manuscript.*

We agree that the findings of Saiepour 2015 are of great importance and that our results confirm their earlier study. Using optogenetics and Dreadds to inactivate PV+ cells in V1 they have revealed that the activity of these cells is not necessary for the expression of the adult OD shift. It is heart-warming to us that these two studies, using different recording methods, have arrived at the same conclusion in regards to the expression mechanisms involved in adult OD plasticity, which only adds to the striking contrast with SRP expression mechanisms that we describe in our manuscript. We now devote a considerably larger portion of our Discussion section to this important study and its implications.